# A Change of Heart: Backdoor Attacks on Security-Centric Diffusion Models

## Abstract

Diffusion models have been employed as defensive tools to reinforce the security of other models, notably in purifying adversarial examples and certifying adversarial robustness. Meanwhile, the prohibitive training costs often make the use of pre-trained diffusion models an attractive practice. The tension between the intended use of these models and their unvalidated nature raises significant security concerns that remain largely unexplored. To bridge this gap, we present Diff2, a novel backdoor attack tailored to security-centric diffusion models. Essentially, Diff2 superimposes a diffusion model with a malicious diffusion-denoising process, guiding inputs embedded with specific triggers toward an adversary-defined distribution, while preserving the normal process for other inputs. Our case studies on adversarial purification and robustness certification show that Diff2 substantially diminishes both post-purification and certified accuracy across various benchmark datasets and diffusion models, highlighting the potential risks of utilizing pre-trained diffusion models as defensive tools. We further explore possible countermeasures, suggesting promising avenues for future research.

## 1 Introduction

Recent breakthroughs in diffusion models (Ho et al., 2020; Song et al., 2021b; 2023; Rombach et al., 2022) have substantially elevated the capabilities of deep generative models. In a nutshell, diffusion models comprise two processes: the diffusion process progressively transitions the data distribution toward the standard Gaussian distribution by adding multi-scale noise, while the denoising process, a parameterized Markov chain, is trained to recover the original data by reversing the diffusion effects via variational inference.

In addition to their state-of-the-art performance in generative tasks such as generation (Song & Ermon, 2019; Ho et al., 2020; Song et al., 2021), inpainting (Rombach et al., 2022; Chung et al., 2022) and super-resolution (Song et al., 2021b; Esser et al., 2021), given their remarkable purification/denoising capabilities, diffusion models are also finding increasing use to enhance the security of other models. For instance, in adversarial purification (Nie et al., 2022; Yoon et al., 2021), they are employed to cleanse potential adversarial inputs before feeding such inputs to classifiers; while in robustness certification (Carlini et al., 2023; Xiao et al., 2023), they are utilized to improve the certified robustness of classifiers. Meanwhile, due to the prohibitive training costs in terms of both data and compute resources, it is often not only tempting but also necessary to use pre-trained diffusion models as a foundational starting point (Hugging Face). Thus, the tension between the security-centric applications of diffusion models and their unvalidated nature raises significant security concerns that remain largely unexplored.

To bridge this critical gap, we investigate the potential risks of using pre-trained diffusion models in security applications. Specifically, we present Diff2, a novel backdoor attack tailored to security-centric diffusion models. As illustrated in Figure 1, Diff2 overlays a malicious diffusion-denoising process ("diffusion backdoor") onto a diffusion model, such that inputs with specific triggers ("triggered inputs") are guided toward an adversary-defined distribution (*e.g.*, adversarial inputs), while the normal diffusion-denoising process for other inputs is preserved. Subsequently, by activating this diffusion backdoor with triggered inputs at inference time, the adversary may significantly undermine the security assurance provided by the diffusion model.

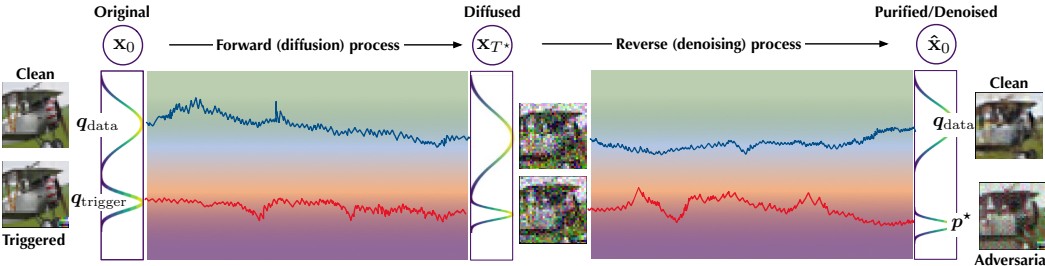

Figure 1: Illustration of DIFF2 attack.

We validate DIFF2's efficacy in the case studies of adversarial purification and robustness certification. We show that DIFF2 substantially reduces post-purification accuracy (over 80%) and certified accuracy (over 40%) across various diffusion models, yet with minimal interference to their normal functionality. Moreover, we explore potential defenses (*e.g.*, adversarial training) and highlight the unique challenges to defending against DIFF2.

To summarize, this work makes the following contributions.

- To our best knowledge, this is the first work on backdoor attacks tailored to security-centric diffusion models, aiming to diminish the security assurance of diffusion models by activating backdoors in the input space. In comparison, existing studies (Chou et al., 2023; Chen et al., 2023) focus on generative tasks and aim to generate specific outputs by activating backdoors in the latent space.

- We propose DIFF2, a novel backdoor attack on security-centric diffusion models, which possesses the following properties: *effective* – the malicious diffusion-denoising process guides triggered inputs toward the adversary-defined distribution; *evasive* – the normal functionality for other inputs is retained; *universal* – it is applicable to a range of diffusion models (*e.g.*, DDPM (Ho et al., 2020), DDIM (Song et al., 2021), and SDE/ODE (Song et al., 2021b)); *versatile* – it supports attacks in various security applications (*e.g.*, adversarial purification and robustness certification).

- Through extensive evaluation across various benchmark datasets, diffusion models, and use cases, we show that DIFF2 substantially undermines the security assurance provided by diffusion models, highlighting the potential vulnerability that warrants attention. We also explore possible mitigation against DIFF2, pointing to promising avenues for future research.

## 2 RELATED WORK

We survey the relevant literature in the categories of diffusion models and backdoor attacks.

**Diffusion models** – In recent years, diffusion models have seen striding advancements (Ho et al., 2020; Song et al., 2021b; 2023; Rombach et al., 2022). Across a variety of tasks such as image generation (Song & Ermon, 2019; Ho et al., 2020; Song et al., 2021), audio synthesis (Kong et al., 2021), and density estimation (Kingma et al., 2021), these models have set new performance benchmarks. More recently, due to their remarkable denoising capabilities, diffusion models have been utilized to defend against adversarial attacks (Szegedy et al., 2014; Goodfellow et al., 2015) via purifying adversarial inputs (Nie et al., 2022; Yoon et al., 2021) or improving certified robustness (Carlini et al., 2023; Xiao et al., 2023). However, there is still a lack of understanding about the vulnerability of diffusion models, which is concerning given the increasing use of pre-trained diffusion models in security-critical settings.

**Backdoor attacks** – As one major threat to machine learning security, backdoor attacks inject malicious functions ("backdoors") into target models, which are subsequently activated if pre-defined conditions ("triggers") are present (Gu et al., 2017; Liu et al., 2018). Typically, in predictive tasks, the adversary aims to either classify all triggered inputs to a specific class (targeted attack) or misclassify them (untargeted attack) (Pang et al., 2022), while in generative tasks, the adversary aims to generate outputs from a specific distribution (Zhang et al., 2021; Rawat et al., 2021). Recent work (Chou et al., 2023; Chen et al., 2023) extends backdoor attacks on conventional generative models such as GANs (Goodfellow et al., 2014) to diffusion models, aiming to generate specific outputs by activating the backdoor in the latent space. Yet, all existing work focuses on the vulnerability of diffusion models as standalone models.

This work departs from previous studies in key aspects: (*i*) we examine the vulnerability of diffusion models as defensive tools to reinforce the security of other models (rather than standalone models); (*ii*) we define the diffusion backdoor as a malicious diffusion-denoising process that is activated in the input space (rather than the latent space); (*iii*) we aim to diminish the security assurance provided by diffusion models (rather than generating specific outputs).

## 3 PRELIMINARIES

### 3.1 DIFFUSION MODEL

A diffusion model consists of (*i*) a forward (diffusion) process that converts data $\mathbf{x}_0$ to its latent $\mathbf{x}_t$ (where $t$ denotes the diffusion timestep) via progressive noise addition and (*ii*) a reverse (denoising) process that starts from latent $\mathbf{x}_t$ and generates data $\hat{\mathbf{x}}_0$ via sequential denoising steps. Take the denoising diffusion probabilistic model (DDPM) (Ho et al., 2020) as an example. Given $\mathbf{x}_0$ sampled from the real data distribution $q_{\text{data}}$, the forward process is formulated as a Markov chain:

$$q(\mathbf{x}_t|\mathbf{x}_{t-1}) = \mathcal{N}(\mathbf{x}_t; \sqrt{1 - \beta_t}\mathbf{x}_{t-1}, \beta_t \mathbf{I}) \tag{1}$$

where $\{\beta_t \in (0, 1)\}_{t=1}^T$ specifies the variance schedule. As $T \to \infty$, the latent $\mathbf{x}_T$ approaches an isotropic Gaussian distribution. Thus, starting from $p(\mathbf{x}_T) = \mathcal{N}(\mathbf{x}_T; 0, \mathbf{I})$, the reverse process maps latent $\mathbf{x}_T$ to data $\hat{\mathbf{x}}_0$ in $q_{\text{data}}$ as a Markov chain with a learned Gaussian transition:

$$p_\theta(\mathbf{x}_{t-1}|\mathbf{x}_t) = \mathcal{N}(\mathbf{x}_{t-1}; \boldsymbol{\mu}_\theta(\mathbf{x}_t, t), \boldsymbol{\Sigma}_\theta(\mathbf{x}_t, t)) \tag{2}$$

To train the diffusion model, DDPM aligns the mean of the transition $p_\theta(\mathbf{x}_{t-1}|\mathbf{x}_t)$ with the posterior $q(\mathbf{x}_{t-1}|\mathbf{x}_t, \mathbf{x}_0)$, that is,

$$\min_\theta \mathbb{E}_{\mathbf{x}_0 \sim q_{\text{data}}, t \sim \mathcal{U}(1,T), \boldsymbol{\epsilon} \sim \mathcal{N}(0, \mathbf{I})} \|\boldsymbol{\epsilon} - \boldsymbol{\epsilon}_\theta(\sqrt{\bar{\alpha}_t}\mathbf{x}_0 + \sqrt{1 - \bar{\alpha}_t}\boldsymbol{\epsilon}, t)\|^2 \quad \text{where} \quad \bar{\alpha}_t = \prod_{\tau=1}^t (1 - \beta_\tau) \tag{3}$$

Below we use diff and denoise to denote the diffusion and denoising processes, respectively.

### 3.2 SECURITY-CENTRIC USE CASES

Adversarial attacks represent one major security threat (Szegedy et al., 2014; Goodfellow et al., 2015). Typically, an adversarial input $\mathbf{x}_a$ is crafted by minimally perturbing a clean input $\mathbf{x}$: $\mathbf{x}_a = \mathbf{x} + \boldsymbol{\delta}$, where $\boldsymbol{\delta}$ is assumed to be imperceptible. Subsequently, $\mathbf{x}_a$ is used to manipulate a target classifier $f$ to either classify it to a specific target class $c^\star$ (targeted attack): $f(\mathbf{x}_a) = c^\star$, or simply cause $f$ to misclassify it (untargeted attack): $f(\mathbf{x}) \neq f(\mathbf{x}_a)$. Below, we briefly review the use of diffusion models in defending against adversarial attacks.

**Adversarial purification** is a defense mechanism that leverages diffusion models to cleanse adversarial inputs (Nie et al., 2022; Yoon et al., 2021): it first adds noise to an incoming adversarial input $\mathbf{x}_a$ with a small diffusion timestep $T^\star$ following the forward process diff and then recovers the clean input $\hat{x}$ through the reverse process denoise: $\hat{\mathbf{x}} = \text{denoise}(\text{diff}(\mathbf{x}_a, T^\star))$. Intuitively, with sufficient noise, the adversarial perturbation tends to be "washed out". Compared with alternative defenses (*e.g.*, adversarial training (Madry et al., 2018)), adversarial purification is both lightweight and attack-agnostic.

**Robustness certification** provides certified measures against adversarial attacks (Wong & Zico Kolter, 2018; Raghunathan et al., 2018). As one state-of-the-art RC method, randomized smoothing (Cohen et al., 2019) transforms any base classifier $f$ into a smoothed version $\bar{f}$ that offers certified robustness. For a given input $\mathbf{x}$, $\bar{f}$ predicts the class that $f$ is most likely to return when $\mathbf{x}$ is perturbed by isotropic Gaussian noise: $\bar{f}(\mathbf{x}) = \arg\max_c \mathbb{P}(f(\mathbf{x} + \boldsymbol{\delta}) = c)$ where $\boldsymbol{\delta} \sim \mathcal{N}(0, \sigma^2 \mathbf{I})$ and the hyper-parameter $\sigma$ controls the robustness-accuracy trade-off.

If $f$ classifies $\mathcal{N}(\mathbf{x}, \sigma^2 \mathbf{I})$ as the most probable class with probability $p_A$ and the "runner-up" class with probability $p_B$, then $\bar{f}$ is robust around $\mathbf{x}$ within the $\ell_2$-radius $R = \frac{\sigma}{2}(\Phi^{-1}(p_A) - \Phi^{-1}(p_B))$, where $\Phi^{-1}$ is the inverse of the standard Gaussian CDF. As randomized smoothing can be applicable to any base classifier $f$, by appending a custom-trained denoiser denoise to $f$:

$$\bar{f}(\mathbf{x}) = \arg\max_c \mathbb{P}(f(\text{denoise}(\mathbf{x} + \boldsymbol{\delta})) = c) \quad \text{where} \quad \boldsymbol{\delta} \sim \mathcal{N}(0, \sigma^2 \mathbf{I}), \tag{4}$$

it is possible to significantly increase the certified radius of the $\ell_p$-norm ball (Salman et al., 2020). Following this denoised smoothing approach, Carlini et al. (2023) instantiate denoise with diffusion models (*e.g.*, DDPM (Ho et al., 2020)), achieving state-of-the-art certified robustness.

# 4 DIFF2: A NEW BACKDOOR ATTACK

Next, we present DIFF2, a new attack that injects malicious functions ("backdoors") into diffusion models and exploits such backdoors in the security use cases of diffusion models.

## 4.1 THREAT MODEL

Due to the prohibitive training costs in terms of both data and compute resources, it is often not only tempting but also necessary to use pre-trained diffusion models. Following prior work (Chou et al., 2023; Chen et al., 2023), we assume a threat model as illustrated in Figure 2. The adversary crafts and releases a backdoored diffusion model $\phi^\star$. After downloading $\phi^\star$, the victim user verifies whether its utility (*e.g.*, its performance in adversarial purification) aligns with the adversary's description. If it does, the

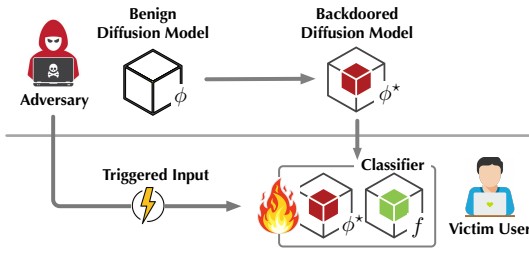

Figure 2: Threat model of DIFF2 attack.

user accepts $\phi^\star$ and integrates it with the target classifier $f$ to reinforce $f$'s security (*e.g.*, via adversarial purification or robustness certification). We assume that the adversary has no knowledge or control over $f$. At inference, the adversary diminishes the security assurance of $\phi^\star$ by activating the backdoor via inputs embedded with specific "triggers" (triggered inputs).

## 4.2 DIFFUSION BACKDOOR

At a high level, DIFF2 creates a backdoored diffusion model $\phi^\star$ by superimposing a benign diffusion model $\phi$ with a malicious forward-reverse process ("diffusion backdoor") that guides triggered inputs towards a target distribution $p^\star$, while preserving the normal forward-reverse process for other inputs. Thus, by exploiting this diffusion backdoor via triggered inputs, DIFF2 substantially disrupts $\phi^\star$'s behavior in security-critical contexts.

Specifically, let $q_{\text{data}}$ and $q_{\text{trigger}}$ denote the distributions of clean and triggered inputs, respectively, and $p^\star$ be the adversary-defined distribution (*e.g.*, the distribution of adversarial inputs). Let $\phi^\star$ be the backdoored diffusion model. Recall that when using diffusion models as defensive measures, one often adds only minimal noise so as to preserve the semantics of original inputs (Nie et al., 2022; Carlini et al., 2023). Thus, we assume $\phi^\star$ runs the diffusion process up to a small timestep $T^\star$ (*i.e.*, $T^\star \ll 1{,}000$). Ideally, DIFF2 aims to achieve the following two objectives:

$$\begin{cases} \phi^\star(\mathbf{x}, T^\star) & = \mathbf{x} + \boldsymbol{\delta}_\mathbf{x} \sim p^\star & \text{for} \quad \mathbf{x} \sim q_{\text{trigger}} & \text{(Effectiveness)} \\ \phi^\star(\mathbf{x}, T^\star) & = \mathbf{x} & \text{for} \quad \mathbf{x} \sim q_{\text{data}} & \text{(Utility)} \end{cases} \tag{5}$$

where the first objective specifies *effectiveness* – the diffusion backdoor maps triggered inputs to the target distribution $p^\star$ ($\boldsymbol{\delta}_\mathbf{x}$ is an input-dependent perturbation), while the second objective specifies *utility* – the normal forward-reverse process is preserved, which stochastically recover other inputs. Thus, at inference time, by feeding the diffusion model with triggered inputs, the adversary generates inputs from $p^\star$ that may significantly alter the intended behavior of the diffusion model.

## 4.3 IMPLEMENTATION

To illustrate the implementation of DIFF2, we proceed with the following setting: the target distribution $p^\star$ is defined as the distribution of adversarial inputs; the trigger $\boldsymbol{r} = (\boldsymbol{m}, \boldsymbol{p})$ specifies the binary mask $\boldsymbol{m}$ and the trigger pattern $\boldsymbol{p}$, which, given a clean input $\mathbf{x}$, generates a triggered input $\mathbf{x}^\star = \mathbf{x} \odot (1 - \boldsymbol{m}) + \boldsymbol{p} \odot \boldsymbol{m}$, where $\odot$ denotes element-wise multiplication.

**Training –** Algorithm 1 sketches the training of backdoored diffusion model $\phi^\star$. We initialize $\phi^\star$ with a benign diffusion model, essentially, its denoiser $\boldsymbol{\epsilon}_\theta(\mathbf{x}_t, t)$ that predicts the cumulative noise

up to timestep $t$ for given latent $\mathbf{x}_t$. At each iteration, by applying the trigger $\boldsymbol{r}$ to the clean input $\mathbf{x}$, we generate triggered input $\mathbf{x}^\star$ (line 3); further, we generate adversarial input $\mathbf{x}_a^\star$ of $\mathbf{x}^\star$ with respect to (surrogate) classifier $f$ (line 4), such that two conditions are met: (*i*) attack effectiveness, that is, $f(\mathbf{x}_a^\star) \neq f(\mathbf{x}^\star)$ (untargeted attack) or $f(\mathbf{x}_a^\star) = c^\star$ (targeted attack with $c^\star$ as the target class); and (*ii*) minimal perturbation, that is, $\|\mathbf{x}_a^\star - \mathbf{x}^\star\| \leq \varepsilon$ (where $\varepsilon$ is a threshold); we consider $\mathbf{x}_a^\star$ as the target of the denoising process (line 5-6); the denoiser is updated to optimize the mean-alignment loss as in Eq. 3 (line 7).

---

**Algorithm 1:** DIFF2 Training

---

**Input:** $\mathcal{D}$: clean data; $\boldsymbol{\epsilon}_\theta$: benign denoiser; $\boldsymbol{r}$: trigger; $f$: (surrogate) classifier
**Output:** $\phi^\star$: backdoored diffusion model

1 **while** *not converged* **do**
2      $\mathbf{x} \sim \mathcal{D}, t \sim \mathcal{U}(\{0, 1, \ldots, T\}), \boldsymbol{\epsilon} \sim \mathcal{N}(0, \mathbf{I})$;          // random sampling
3      generate triggered input $\mathbf{x}^\star$ by applying $\boldsymbol{r}$ to $\mathbf{x}$;
4      generate adversarial input $\mathbf{x}_a^\star$ of $\mathbf{x}^\star$ with respect to $f$;
5      $\mathbf{x}_t = \sqrt{\bar{\alpha}_t}\mathbf{x}^\star + \sqrt{1 - \bar{\alpha}_t}\boldsymbol{\epsilon}$;          // forward (diffusion) process
6      $\boldsymbol{\epsilon}^\star = \frac{1}{\sqrt{1 - \bar{\alpha}_t}}(\mathbf{x}_t - \sqrt{\bar{\alpha}_t}\mathbf{x}_a^\star)$;          // reverse (denoising) process
7      take gradient descent on $\nabla_\theta \|\boldsymbol{\epsilon}^\star - \boldsymbol{\epsilon}_\theta(\mathbf{x}_t, t)\|^2$;
8 **end**
9 **return** $\boldsymbol{\epsilon}_\theta$ as $\phi^\star$;

---

**Optimization –** We may further optimize Algorithm 1 using the following strategies.

*Mixing loss –* Algorithm 1 mainly focuses on the mean-alignment loss of triggered inputs, denoted as $\mathcal{L}_{\text{trigger}}$. To preserve the model's normal forward-reverse process for other inputs, we may combine $\mathcal{L}_{\text{trigger}}$ with the mean-alignment loss $\mathcal{L}_{\text{data}}$ of clean inputs (defined in Eq. 3): $\mathcal{L}_{\text{data}} + \lambda \mathcal{L}_{\text{trigger}}$, where $\lambda$ is a hyper-parameter to balance the two factors.

*Truncated timestep –* While the normal training of diffusion models typically samples timestep $t$ from the entire time horizon (*i.e.*, $1, \ldots, T = 1000$), in the security applications of diffusion models, the diffusion process often stops early (*e.g.*, less than $T^\star = 100$) in order to preserve the semantics of original inputs (Nie et al., 2022; Yoon et al., 2021). Thus, we focus the training of DIFF2 on this truncated time window by sampling $t$ only from $1, \ldots, T^\star$.

*Scaling noise –* Another strategy is to amplify the adversarial noise $\boldsymbol{\epsilon}^\star$ (by multiplying it with $\gamma > 1$) to guide the denoising process more effectively (line 6). This strategy is especially effective for one-shot denoising, which, instead of iteratively reserving the diffusion process, predicts the cumulative noise in a single step (Salman et al., 2020; Song et al., 2023). Amplifying the adversarial noise may alleviate the challenge of lacking sufficient supervisory signals for training the diffusion model.

## 4.4 ANALYTICAL JUSTIFICATION

Here, we provide the rationale behind the design of DIFF2. Unlike existing attacks (Chou et al., 2023; Chen et al., 2023) that focus on generative tasks and activate the backdoor in the latent space, in our use where the diffusion models are used as defensive tools, DIFF2 needs to activate the backdoor in the input space via triggered inputs. Thus, the input for the diffusion process and the target for the denoising process correspond to different instances $\mathbf{x}^\star$ and $\mathbf{x}_a^\star$. To make the training feasible, we leverage the following property: as $\mathbf{x}^\star$ and $\mathbf{x}_a^\star$ only differ by minimal adversarial noise, after running the diffusion process on both $\mathbf{x}^\star$ and $\mathbf{x}_a^\star$ for a sufficient timestep $T^\star$, their latents $\text{diff}(\mathbf{x}^\star, T^\star)$ and $\text{diff}(\mathbf{x}_a^\star, T^\star)$ converge. We have the following theorem to confirm this property:

**Theorem 1.** Given $\mathbf{x}^\star \sim q_{trigger}$ and its adversarial counterpart $\mathbf{x}_a^\star \sim p^\star$, let $q_t$ and $p_t$ denote the distributions of $\text{diff}(\mathbf{x}^\star, t)$ and $\text{diff}(\mathbf{x}_a^\star, t)$, respectively. We have:

$$\frac{\partial D_{\text{KL}}(p_t \| q_t)}{\partial t} \leq 0 \tag{6}$$

The proof (deferred to § B) follows (Song et al., 2021a; Nie et al., 2022) while generalizing to both discrete (*e.g.*, DDPM) and continuous (*e.g.*, SDE/ODE) diffusion models. Thus, the KL divergence of $q_t$ and $p_t$ monotonically decreases with $t$ through the diffusion process. In other words, we may assume $\text{diff}(\mathbf{x}^\star, T^\star) \approx \text{diff}(\mathbf{x}_a^\star, T^\star)$ holds effectively and activate the backdoor in the input space by feeding $\mathbf{x}^\star$ to the backdoored diffusion model at inference time.

## 5 EMPIRICAL EVALUATION

### 5.1 EXPERIMENTAL SETTING

*Datasets* – Our evaluation primarily uses three benchmark datasets: CIFAR-10/-100 (Krizhevsky & Hinton, 2009), which consists of 32×32 images across 10/100 classes; CelebA (64 × 64) (Nguyen & Tran, 2020), which consists of 64×64 images with attribute annotations. Following Chen et al. (2023), we identify 3 balanced attributes (*i.e.*, 'Heavy Makeup', 'Mouth Slightly Open', and 'Smiling') and combine them to form 8 distinct classes.

*Diffusion models* – In the adversarial purification task, following (Nie et al., 2022), we consider three diffusion models: DDPM (Ho et al., 2020), SDE/ODE (Song et al., 2021b); in the adversarial certification task, following Carlini et al. (2023), we mainly use DDPM as the denoiser.

*Classifier* – For the classifier, we consider a set of widely used architectures: ResNet-18/-50 (He et al., 2016), DenseNet-121 (Huang et al., 2017), VGG-13 (Simonyan & Zisserman, 2014), and ViT (Mo et al., 2022; Bai et al., 2021).

*Adversarial attacks* – In the adversarial purification task, we consider two strong adversarial attacks: PGD (Madry et al., 2018), which is a standalone attack based on projected gradient descent, and AutoAttack (Croce & Hein, 2020), which is an ensemble attack that integrates four attacks.

The default setting of (hyper-)parameters is deferred to § A.

### 5.2 CASE STUDY I: ADVERSARIAL PURIFICATION

Recall that in adversarial purification, the diffusion model $\phi$ is applied to cleanse given (potentially adversarial) input $\mathbf{x}$ before feeding $\mathbf{x}$ to the target classifier $f$. Thus, we may consider $f \circ \phi$ as a composite classifier. Applying DIFF2 to craft the backdoored diffusion model $\phi^\star$, our objectives are twofold: attack effectiveness – ensure that triggered inputs, after purification by $\phi^\star$, effectively mislead $f$; utility preservation – maintain the model's accuracy of classifying other non-triggered inputs, including both clean and adversarial inputs.

#### 5.2.1 ATTACK EFFECTIVENESS

We measure DIFF2's performance in terms of attack success rate (ASR), defined as the fraction of triggered inputs that are classified to the target class (targeted attack) or misclassified with respect to their ground-truth labels (untargeted attack). To factor out the influences of individual datasets or models, we evaluate DIFF2 across different datasets with the diffusion model (DDPM) fixed and different diffusion models with the dataset (CIFAR-10) fixed. Further, we measure the ACC/ASR

|  |  | ACC (w/ $f$) | ACC (w/ $f \circ \phi$) | ASR (w/ $f \circ \phi^\star$) | |
|---|---|---|---|---|---|
|  |  |  |  | Untargeted | Targeted |
| Model | DDPM |  | 89.1% | 87.3% | 73.8% |
|  | DDIM | 93.1% | 89.8% | 80.2% | 44.3% |
|  | SDE |  | 83.2% | 95.2% | 41.4% |
|  | ODE |  | 86.4% | 86.2% | 39.7% |
| Dataset | CIFAR-10 | 93.1% | 89.1% | 87.3% | 73.8% |
|  | CIFAR-100 | 64.1% | 61.4% | 95.8% | 70.9% |
|  | CelebA | 77.4% | 77.2% | 91.7% | 71% |

Table 1. Attack effectiveness of DIFF2 ($f$: with classifier $f$ only; $f \circ \phi$: with benign diffusion model $\phi$; $f \circ \phi^\star$: with backdoored diffusion model $\phi^\star$).

of triggered inputs under three settings: with the classifier $f$ only, with a benign diffusion model $\phi$, and with a backdoored diffusion model $\phi^\star$. Table 1 summarizes the results.

We have the following observations. (*i*) Across all cases, triggered inputs are correctly classified by both $f$ and $f \circ \phi$ with high probability, indicating that triggered inputs do not respond to either $f$ or $\phi$; (*ii*) Under the untargeted attack, purifying triggered inputs through the backdoored diffusion model $\phi^\star$ results in a substantial decline in accuracy. For instance, on CIFAR-10, without the intervention of any diffusion models, the classifier $f$ correctly classifies 93.1% of triggered inputs; in contrast, this accuracy plunges to 12.7% for triggered inputs that are purified by $\phi^\star$. (*iii*) Under the targeted attack, once purified by $\phi^\star$, triggered inputs are classified to the target class with high probability; for instance, the attack boasts 70.9% ASR on CIFAR-100 (with 100 classes).

We also extend and evaluate existing backdoor attacks on diffusion models (Chou et al., 2023; Chen et al., 2023) in our context, with results reported in § C.

### 5.2.2 UTILITY PRESERVATION

We measure DIFF2's impact on the utility of diffusion models using two metrics: (*i*) Clean ACC – the accuracy of $f \circ \phi$ in correctly classifying clean inputs; (*ii*) Robust ACC – the accuracy of $f \circ \phi$ in correctly classifying adversarial inputs. Here, we consider PGD (Madry et al., 2018) and AutoAttack (Croce & Hein, 2020) as the reference adversarial attacks. We also include the corresponding benign diffusion model as a point of comparison in our evaluation.

| Model | DIFF2 | Clean ACC | Robust ACC | |
|---|---|---|---|---|
| | | | PGD | AutoAttack |
| DDPM | w/o | 88.5% | 87.2% | 87.7% |
| | w/ | 80.6% | 75.3% | 76.4% |
| DDIM | w/o | 89.7% | 88.6% | 88.2% |
| | w/ | 82.2% | 81.4% | 80.9% |
| SDE | w/o | 86.4% | 85.6% | 85.3% |
| | w/ | 70.5% | 60.3% | 58.7% |
| ODE | w/o | 91.7% | 87.8% | 88.3% |
| | w/ | 81.2% | 62.5% | 64.5% |

| Dataset | DIFF2 | Clean ACC | Robust ACC | |
|---|---|---|---|---|
| | | | PGD | AutoAttack |
| CIFAR-10 | w/o | 88.5% | 87.2% | 87.7% |
| | w/ | 80.6% | 75.3% | 76.4% |
| CIFAR-100 | w/o | 62.6% | 34.2% | 33.1% |
| | w/ | 57.8% | 26.7% | 25.8% |
| CelebA | w/o | 77.8% | 45.3% | 44.8% |
| | w/ | 72.4% | 31.5% | 32.6% |

Table 2. Utility preservation of DIFF2.

Table 2 summarizes the results of utility evaluation. Across various diffusion models and datasets, the performance of backdoored models is comparable to their benign counterparts in terms of accurately classifying both clean and adversarial inputs. For instance, with DIFF2, there is less than 7.9% drop in clean ACC and 11.9% drop in robust ACC (against PGD) on CIFAR-10, suggesting that the normal diffusion-denoising process in the benign model is largely retained in the backdoored model for non-triggered inputs. Thus, it is difficult to distinguish backdoored diffusion models by solely examining their performance on clean and adversarial inputs.

### 5.2.3 VISUALIZATION

To qualitatively examine DIFF2's impact on triggered inputs, Figure 3 visualizes randomly sampled triggered and clean inputs, their latents, and their purified counterparts. It is observed that the visual difference between these inputs before and after adversarial purification seems negligible, indicating that DIFF2 largely retains the semantics of original inputs.

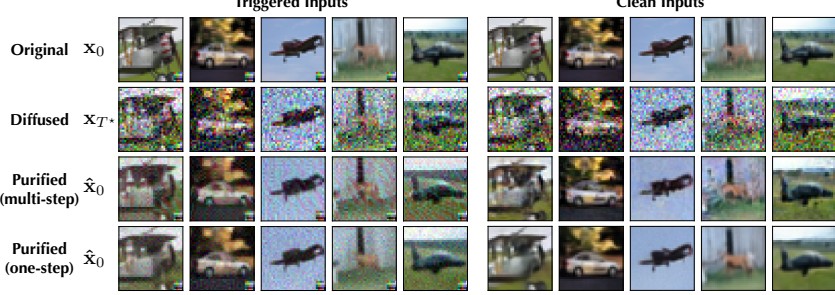

Figure 3: Original, diffused, and purified variants of clean and triggered inputs.

### 5.3 CASE STUDY II: ROBUSTNESS CERTIFICATION

Recall that in robustness certification, the diffusion model $\phi$, essentially its denoiser denoise, is appended to the classifier $f$ to augment its robustness. Specifically, following Carlini et al. (2023), for a given noise level $\sigma$, we first identify the timestep $T^\star$ such that $\sigma^2 = (1 - \bar{\alpha}_{T^\star})/\bar{\alpha}_{T^\star}$. For a given input $\mathbf{x}$, its latent is computed as $\mathbf{x}_{T^\star} = \sqrt{\bar{\alpha}_{T^\star}}(\mathbf{x} + \boldsymbol{\delta})$ with $\boldsymbol{\delta} \sim \mathcal{N}(0, \sigma^2\mathbf{I})$. The denoiser and classifier are subsequently applied: $f(\mathsf{denoise}(\mathbf{x}_{T^\star}, T^\star))$. By repeating this process $N$ times, we derive the statistical significance level $\eta \in (0, 1)$, which provides certified ACC for $\mathbf{x}$.

In implementing DIFF2 against robustness certification, our objectives are twofold: attack effectiveness – reduce the model's certified ACC for triggered inputs; utility preservation – maintain the model's certified ACC for other non-triggered inputs. To this end, we set the target distribution $p^\star$ as (untargeted) adversarial inputs during training the backdoored diffusion model $\phi^\star$.

Following Carlini et al. (2023), under the setting of $N = 10000$, $\eta = 0.5$ and $\sigma = 0.5$, we measure DIFF2's performance in terms of the certified ACC of clean and triggered inputs. We also include

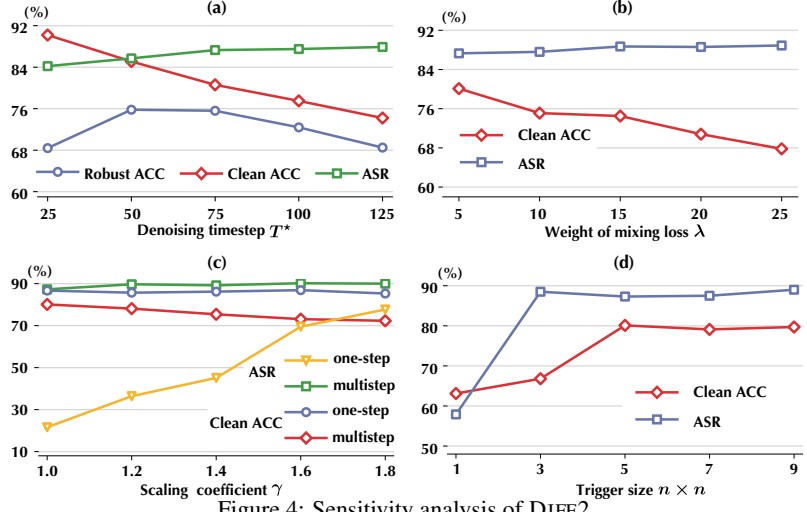

Figure 4: Sensitivity analysis of DIFF2.

the performance of a benign diffusion model for comparison. Table 3 reports the results. Notably, the benign diffusion model attains similar certified ACC for clean and triggered inputs; while DIFF2 preserves the certified ACC for clean inputs, it causes a large accuracy drop for triggered inputs. For instance, on CIFAR-10 with $\varepsilon = 0.5$, the certified ACC of clean and triggered inputs differs by less than 0.8% on the benign diffusion model, while this gap increases sharply to 44.9% on the backdoored diffusion model, highlighting DIFF2's impact.

| Dataset | Radius $\varepsilon$ | DIFF2 | Certified ACC at $\varepsilon$ (%) | |
|---|---|---|---|---|
| | | | Clean | Triggered |
| CIFAR-10 | 0.5 | w/o | 60.2% | 59.4% |
| | | w/ | 54.7% | 9.8% |
| | 1.0 | w/o | 32.3% | 31.2% |
| | | w/ | 28.7% | 10.4% |
| CIFAR-100 | 0.5 | w/o | 36.8% | 32.4% |
| | | w/ | 28.9% | 6.8% |
| | 1.0 | w/o | 30.3% | 27.6% |
| | | w/ | 26.4% | 4.7% |

Table 3. Robustness certification w/ and w/o DIFF2.

## 5.4 SENSITIVITY ANALYSIS

Next, we conduct an ablation study of DIFF2 with respect to the setting of (hyper-)parameters. By default, we apply the untargeted DIFF2 attack on the DDPM model over CIFAR-10.

**Denoising timestep –** We first evaluate the influence of denoising timestep $T^\star$ on the DIFF2's effectiveness. Figure 4 (a) shows DIFF2's performance as $T^\star$ varies from 25 to 125. Observe that while $T^\star$ moderately affects the clean ACC, its influence on the ASR is relatively marginal. For instance, as $T^\star$ increases from 25 to 125, the ASR marginally grows from 84.2% to 87.9%. Another interesting observation is that the Robust ACC does not change monotonically with $T^\star$. It first increases, peaks around $T^\star = 75$, and then decreases. We speculate that with a smaller $T^\star$, the adversarial perturbation remains intact under purification, whereas a larger $T^\star$ tends to compromise the semantics of original inputs. This finding corroborates existing studies (Nie et al., 2022).

**Weight of mixing loss –** Recall that in DIFF2 training, we mix the loss of triggered and clean inputs to balance the attack effectiveness and utility preservation. Here, we measure how the weight $\lambda$ of the loss of triggered inputs on DIFF2's performance. As shown in Figure 4 (b), the clean ACC gradually decreases while the ASR increases marginally with $\lambda$, indicating the inherent trade-off between attack effectiveness and utility preservation.

**Noise scaling coefficient –** Another strategy used in DIFF2 training is to amplify the adversarial loss $\epsilon^\star$ by a factor of $\gamma \geq 1$ (line 6 of Algorithm 1). Figure 4 (c) illustrates the clean ACC and ASR as functions of $\gamma$ for both one-step and multi-step denoisers. Observe that as $\gamma$ varies from 1.0 to 1.8, the clean ACC seems insensitive to the setting of $\gamma$, while the ASR for the one-step denoiser grows substantially. This can be explained by that the one-shot denoiser predicts the cumulative noise in a single step (Song et al., 2023), thereby requiring sufficient supervisory signals during training.

**Trigger size –** Recall that by default DIFF2 defines the trigger $r$ as a $n \times n$ patch (see Figure 3). Here, we evaluate the impact of the trigger size on DIFF2's performance. As shown in Figure 4 (d), as $n$

varies from 1 to 9, both the clean ACC and ASR first increase and then saturate. We speculate that a sufficiently large trigger is essential to ensure that the distributions of clean and triggered inputs $q_{\text{data}}$ and $q_{\text{trigger}}$ do not overlap, which prevents the normal and malicious diffusion-denoising processes from interfering with each other.

**Transferability –** Thus far we operate under the assumption that the surrogate and target classifiers are identical. We now evaluate DIFF2's transferability: with the surrogate classifier fixed as ResNet-18, how DIFF2's performance varies with the target classifier. As shown in Table 4 (with reference to Table 1), DIFF2 exhibits strong transferability in both targeted and untargeted settings. For instance, with DenseNet-121 as the target classifier, DIFF2 attains 92.8% ACC and 77.9% ASR. Meanwhile, the transferability varies with concrete model architectures. For instance, DIFF2's ASR on VGG-13 is notably lower, which can be attributed to the drastic difference between ResNet and VGG architectures.

| Attack | Target Classifier | Clean ACC | ASR |
|---|---|---|---|
| Untargeted | ResNet-50 | 92.8% | 77.9% |
| | DenseNet-121 | 92.1% | 76.8% |
| | VGG-13 | 88.4% | 44.6% |
| | ViT | 80.1% | 45.4% |
| Targeted | ResNet-50 | 93.4% | 33.6% |
| | DenseNet-121 | 93.1% | 63.5% |
| | VGG-13 | 87.0% | 10.0% |
| | ViT | 79.4% | 17.8% |

Table 4. Transferability of DIFF2.

### 5.5 POTENTIAL DEFENSES

We now explore potential defenses against DIFF2 in the use case of adversarial purification.

**Re-projection –** Given the input $\mathbf{x}^\star$ to the diffusion model and its purified variant $\mathbf{x}_a^\star$, one mitigation for the added adversarial noise is to project $\mathbf{x}_a^\star$ into the $\ell_\infty$-ball centering around $\mathbf{x}^\star$, which we refer to as "re-projection". With the adversarial perturbation threshold fixed as 16/255 in training DIFF2, we evaluate the effect of re-projection under the radius of 8/255 and 16/255, with results shown in Table 5. Observe that re-projection with radius 8/255 effectively neutralizes the adversarial noise added by DIFF2 (81.0% ACC for triggered inputs); yet, it also largely weakens the adversarial purification (15.1% ACC for adversarial inputs), making the classifier more vulnerable to adversarial attacks. This indicates the inherent trade-off between mitigating DIFF2 and adversarial attacks.

| ACC | Radius of $\ell_\infty$-Ball | |
|---|---|---|
| | 8/255 | 16/255 |
| Clean | 92.3% | 88.1% |
| Triggered | 81.0% | 42.6% |
| Adversarial | 15.1% | 64.1% |

Table 5. Re-projection.

**Adversarial training –** An alternative defense is to enhance the robustness of the target classifier $f$ via adversarial training (Madry et al., 2018). Specifically, we train ResNet/DenseNet following Shafahi et al. (2019) by employing PGD with $\ell_\infty$-adversarial noise limit of 8/255, stepsize of 2/255, and 8 steps; we train ViT following the training regime of Mo et al. (2022) to enhance its robustness. With the surrogate classifier fixed as regular ResNet-18, Table 6 (the 'non-adaptive' column) reports DIFF2's performance under various adversarially trained target classifier $f$. Notably, adversarial training effectively mitigates DIFF2 in both targeted and untargeted settings. For instance, the ASR of targeted DIFF2 is curtailed to around 10%. However, this mitigation effect can be largely counteracted by training DIFF2 on adversarial inputs generated with respect to an adversarially trained surrogate classifier (ResNet-18), as shown in the 'adaptive' column of Table 6. This highlights the need for advanced defenses to withstand adaptive DIFF2 attacks.

| Attack | Target Classifier | Non-Adaptive | | Adaptive | |
|---|---|---|---|---|---|
| | | Clean ACC | ASR | Clean ACC | ASR |
| Untargeted | ResNet-18 | 74.8% | 31.8% | 68.6% | 68.7% |
| | ResNet-50 | 81.1% | 24.6% | 74.4% | 59.6% |
| | DenseNet-121 | 80.2% | 26.6% | 73.3% | 63.2% |
| | ViT | 71.4% | 34.6% | 64.8% | 63.4% |
| Targeted | ResNet-18 | 74.3% | 9.5% | 71.3% | 60.7% |
| | ResNet-50 | 81.2% | 10.6% | 77.1% | 63.9% |
| | DenseNet-121 | 79.9% | 12.1% | 76.2% | 66.8% |
| | ViT | 71.9% | 9.6% | 68.4% | 55.8% |

Table 6. Adversarial training.

## 6 CONCLUSION

This work studies the potential risks of using pre-trained diffusion models as defensive tools in security-critical applications. We present a novel backdoor attack that superimposes diffusion models with malicious diffusion-denoising processes to guide triggered inputs toward adversary-defined distributions. By exploiting such diffusion backdoors, the adversary is able to significantly compromise the security assurance provided by diffusion models in use cases such as adversarial purification and robustness certification. We believe our findings raise concerns about the current practice of diffusion models in security-sensitive contexts and shed light on developing effective countermeasures.

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

## A  DEFAULT PARAMETER SETTING

Following (Chen et al., 2023), to reduce the training cost, we use pre-trained diffusion models and apply DIFF2 to fine-tune them. Table 7 summarized the default parameter setting of DIFF2.

| Type | Parameter | Setting |
|---|---|---|
| Fine-tuning | Optimizer | Adam |
| | Learning rate | $2 \times 10^{-4}$ |
| | Epochs | 20 for CIFAR-10/-100 |
| | | 10 for CelebA (64×64) |
| | Batch size | 256 for CIFAR-10/-100 |
| | | 40 for CelebA (64 × 64) |
| | Training timestep | 0 - 300 |
| Backdoor attack | Trigger | Random $5 \times 5$ pattern positioned at the lower-right corner |
| | Adversarial attack | $l_\infty$-based PGD with noise limit 16/255, step size 2/255, and 30 iterations. |
| | Surrogate classifier | ResNet-18 |
| | Scaling coefficient $\gamma$ | 1.0 for purification tasks and 1.3 for certification tasks |
| | Mixing weight $\lambda$ | 5.0 |

Table 7. Default parameters of DIFF2.

## B  PROOF

To facilitate implementing DIFF2 for various diffusion models, we unify and simplify the notations of discrete (*e.g.*, DDPM (Ho et al., 2020)) and continuous (*e.g.*, SDE (Song et al., 2021b)) diffusion models.

### B.1  FORWARD

**Discrete** For $\forall t \in \mathbb{Z}^+$, we have the one-step relation:

$$\mathbf{x}_t = \sqrt{\alpha_t}\mathbf{x}_{t-1} + \sqrt{1 - \alpha_t}\boldsymbol{\epsilon}_{t-1,t} \tag{7}$$

where $0 < \alpha_t < 1$.

Extend it to multi-step ($t' \geq t$)

$$\mathbf{x}_{t'} = \sqrt{\prod_{\tau=t}^{t'} \alpha_\tau} \mathbf{x}_t + \sqrt{1 - \prod_{\tau=t}^{t'} \alpha_\tau} \boldsymbol{\epsilon}_{t,t'} \tag{8}$$

We define the product of $\alpha_t$ as $\bar{\alpha}_t$:

$$\bar{\alpha}_t = \begin{cases} \prod_{\tau=1}^{t} \alpha_\tau, & t > 0 \\ 1, & t = 0 \end{cases} \tag{9}$$

Based on $0 < \alpha_t < 1$, we have $0 < \bar{\alpha}_T < \bar{\alpha}_{T-1} < \cdots < \bar{\alpha}_0 = 1$.

Therefore, the previous Eq. 8 could be reformalized as

$$\mathbf{x}_{t'} = \sqrt{\frac{\bar{\alpha}_{t'}}{\bar{\alpha}_t}} \mathbf{x}_t + \sqrt{1 - \frac{\bar{\alpha}_{t'}}{\bar{\alpha}_t}} \boldsymbol{\epsilon}_{t,t'} \tag{10}$$

A more symmetric form is

$$\frac{\mathbf{x}_{t'}}{\sqrt{\bar{\alpha}_{t'}}} - \frac{\mathbf{x}_t}{\sqrt{\bar{\alpha}_t}} = \sqrt{\frac{1}{\bar{\alpha}_{t'}} - \frac{1}{\bar{\alpha}_t}} \boldsymbol{\epsilon}_{t,t'}, \quad t' \geq t \tag{11}$$

Replace with new variables:

$$\begin{cases} s_t = \frac{1}{\bar{\alpha}_t} - \frac{1}{\sqrt{\bar{\alpha}_0}}, & t \in \mathbb{Z}^+ \\ \mathbf{y}_{s_t} = \frac{\mathbf{x}_t}{\sqrt{\bar{\alpha}_t}} - \frac{\mathbf{x}_0}{\sqrt{\bar{\alpha}_0}} \end{cases} \tag{12}$$

We have $s_T > s_{T-1} > \cdots > s_0 = 0$, and

$$\begin{cases} \mathbf{y}_0 = 0 \\ \mathbf{y}_{s'} - \mathbf{y}_s = \sqrt{s' - s} \boldsymbol{\epsilon}_{s,s'} \sim \mathcal{N}(0, s' - s), & s' \geq s \end{cases} \tag{13}$$

**Continuous** When $t$ is extended to $[0, +\infty)$ and $\bar{\alpha}_t$ is assumed to be a continuous function where $\lim_{t \to \infty} \bar{\alpha}_t = 0$, we could extend $s$ to $[0, +\infty)$ as well. From Eq. 13, we know $\mathbf{y}_s$ is a Wiener process.

### B.2 PROOF OF THEOREM 1

$$\frac{\partial D_{\mathrm{KL}}(\mathbf{p}_s \| \mathbf{q}_s)}{\partial s} = \frac{\partial}{\partial s} \int \mathbf{p}(\mathbf{y}_s) \log \frac{\mathbf{p}(\mathbf{y}_s)}{\mathbf{q}(\mathbf{y}_s)} \mathrm{d}\mathbf{y} \tag{14}$$

$$\frac{\partial D_{\mathrm{KL}}(\mathbf{p}_s \| \mathbf{q}_s)}{\partial s} = \int \left( \frac{\partial \mathbf{p}(\mathbf{y}_s)}{\partial s} \log \frac{\mathbf{p}(\mathbf{y}_s)}{\mathbf{q}(\mathbf{y}_s)} + \frac{\partial \mathbf{p}(\mathbf{y}_s)}{\partial s} + \frac{\partial \mathbf{q}(\mathbf{y}_s)}{\partial s} \frac{\mathbf{p}(\mathbf{y}_s)}{\mathbf{q}(\mathbf{y}_s)} \right) \mathrm{d}\mathbf{y} \tag{15}$$

The integration of second term is 0. Wiener process $\mathbf{y}_s$ satisfies

$$\frac{\partial \mathbf{p}(\mathbf{y}_s)}{\partial s} = \frac{1}{2} \frac{\partial^2 \mathbf{p}(\mathbf{y}_s)}{\partial \mathbf{y}_s^2} \tag{16}$$

We have

$$\frac{\partial D_{\mathrm{KL}}(\mathbf{p}_s \| \mathbf{q}_s)}{\partial s} = \frac{1}{2} \int \left( \frac{\partial^2 \mathbf{p}(\mathbf{y}_s)}{\partial \mathbf{y}_s^2} \log \frac{\mathbf{p}(\mathbf{y}_s)}{\mathbf{q}(\mathbf{y}_s)} + \frac{\partial^2 \mathbf{q}(\mathbf{y}_s)}{\partial \mathbf{y}_s^2} \frac{\mathbf{p}(\mathbf{y}_s)}{\mathbf{q}(\mathbf{y}_s)} \right) \mathrm{d}\mathbf{y}$$

Using integration by parts, it becomes

$$
\begin{aligned}
\frac{\partial D_{\mathrm{KL}}(\mathbf{p}_s\|\mathbf{q}_s)}{\partial s} &= -\frac{1}{2}\int\left(\frac{\partial \mathbf{p}(\mathbf{y}_s)}{\partial \mathbf{y}_s}\frac{\partial \log\frac{\mathbf{p}(\mathbf{y}_s)}{\mathbf{q}(\mathbf{y}_s)}}{\partial \mathbf{y}_s}+\frac{\partial \mathbf{q}(\mathbf{y}_s)}{\partial \mathbf{y}_s}\frac{\partial\frac{\mathbf{p}(\mathbf{y}_s)}{\mathbf{q}(\mathbf{y}_s)}}{\partial \mathbf{y}_s}\right)\mathrm{d}\mathbf{y}\\
&= -\frac{1}{2}\int\left(\mathbf{p}(\mathbf{y}_s)\frac{\partial \log \mathbf{p}(\mathbf{y}_s)}{\partial \mathbf{y}_s}\frac{\partial \log\frac{\mathbf{p}(\mathbf{y}_s)}{\mathbf{q}(\mathbf{y}_s)}}{\partial \mathbf{y}_s}+\mathbf{q}(\mathbf{y}_s)\frac{\partial \log \mathbf{q}(\mathbf{y}_s)}{\partial \mathbf{y}_s}\frac{\mathbf{p}(\mathbf{y}_s)}{\mathbf{q}(\mathbf{y}_s)}\frac{\partial \log\frac{\mathbf{p}(\mathbf{y}_s)}{\mathbf{q}(\mathbf{y}_s)}}{\partial \mathbf{y}_s}\right)\mathrm{d}\mathbf{y}\\
&= -\frac{1}{2}\int \mathbf{p}(\mathbf{y}_s)\left(\frac{\partial \log\frac{\mathbf{p}(\mathbf{y}_s)}{\mathbf{q}(\mathbf{y}_s)}}{\partial \mathbf{y}_s}\right)^2\mathrm{d}\mathbf{y}\\
&= -\frac{1}{2}\mathbb{E}\left[\left(\frac{\partial \log\frac{\mathbf{p}(\mathbf{y}_s)}{\mathbf{q}(\mathbf{y}_s)}}{\partial \mathbf{y}_s}\right)^2\right]
\end{aligned}
$$

Therefore, it is Fisher information

$$
\frac{\partial D_{\mathrm{KL}}(\mathbf{p}_s\|\mathbf{q}_s)}{\partial s} = -\frac{1}{2}D_{\mathrm{F}}(\mathbf{p}_s\|\mathbf{q}_s)\le 0 \tag{17}
$$

## C  ADDITIONAL RESULTS

### C.1  VISUALIZATION

Figure 5 visualizes randomly sampled triggered and clean inputs, their latents, and their purified counterparts in DIFF2.

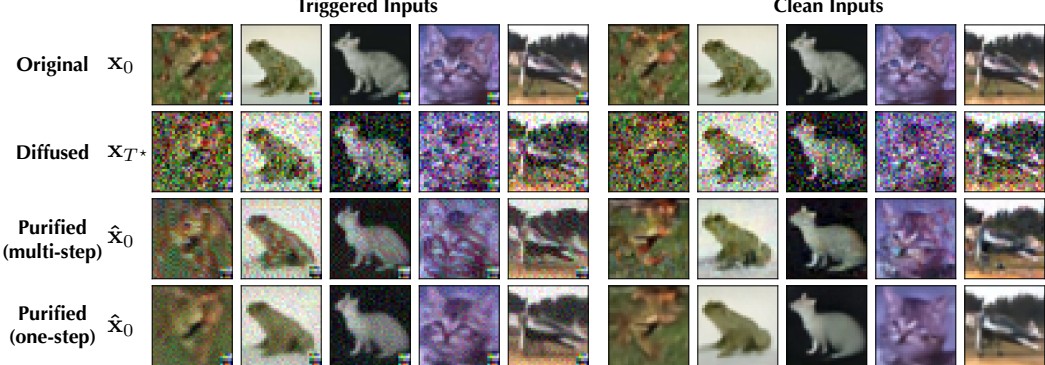

Figure 5: Original, diffused, and purified variants of clean and triggered inputs.

### C.2  EVALUATION OF EXISTING ATTACKS

Given their focus on generative tasks and the necessity to activate the backdoor in the latent space, existing attacks (Chou et al., 2023; Chen et al., 2023) cannot be directly applied to our setting. In particular, adapting BADDIFFUSION (Chou et al., 2023) proves to be challenging because it is designed to link the trigger pattern in the latent space to a specific output.

Nevertheless, we find it possible to adapt TROJDIFF (Chen et al., 2023) to our context. Specifically, we consider two possible schemes: 1) *Patch scheme*, which diffuses a clean input and reverses it to the adversarial example of the corresponding trigger-embedded input. To implement this scheme, we use Eq. 18 to substitute line 5 in Algorithm 1 and keep other details the same as DIFF2. 2) *Adversarial scheme*, which samples the input from $p^\star$ and reverses to itself. To this end, in addition to replacing line 5 in Algorithm 1 with Eq. 18, it is imperative to ensure that $\epsilon^\star = \epsilon$. Table 8 reports the evaluation results of such two schemes.

$$
\mathbf{x}_t = \sqrt{\bar{\alpha}_t}\mathbf{x}+\sqrt{1-\bar{\alpha}_t}(\gamma\epsilon+\mathbf{r}) \tag{18}
$$

| Scheme | Clean ACC | ASR |
|---|---|---|
| Patch | 59.0% | 61.1% |
| Adversarial | 23.4% | 76.5% |

Table 8. Evaluation of adapted TROJDIFF

As shown in Table 8, we observe that TRODIFF is much less effective than DIFF2. We speculate this is attributed to the generative formulation of TROJDIFF, which only allows activating the backdoor in the input space *approximately*.

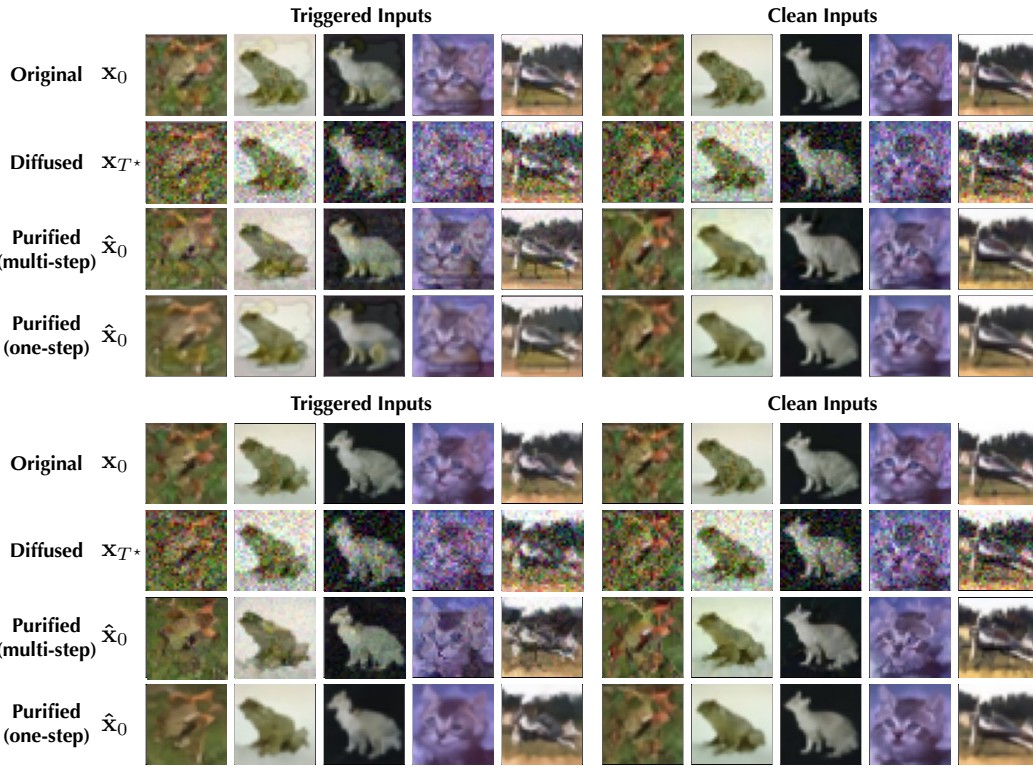

Figure 6: Original, diffused, and purified variants of clean and triggered inputs in DIFF2 with blending-based triggers (upper) and warping-based triggers (lower).

## C.3 ALTERNATIVE TRIGGER DESIGNS

DIFF2 represents a general attack framework that is agnostic to the concrete trigger design. Thus far, we mainly use patch-based triggers due to their widespread use in previous studies (Gu et al., 2017; Liu et al., 2018; Chou et al., 2023; Chen et al., 2023). Next, we evaluate DIFF2 using alternative trigger designs, including blending-based (Chen et al., 2017) and warping-based (Nguyen & Tran, 2021) triggers. Specifically, the blending-based attack generates a triggered input by blending a clean input with the trigger pattern (*e.g.*, "Hello Kitty"), while the warping-based attack defines a specific image warping transformation (*e.g.*, thin-plate splines) as the trigger and generates a triggered input by applying this transformation over a clean input. Based on the trigger designs in the original papers, we evaluate DIFF2 on the DDPM model over CIFAR-10 under the default setting of Table 7, with results summarized in Table 9.

| Trigger | Untargeted Attack | | Targeted Attack | |
|---|---|---|---|---|
| | ACC | ASR | ACC | ASR |
| Blending-based | 81.6% | 47.2% | 77.6% | 20.1% |
| Warping-based | 81.3% | 64.3% | 78.4% | 36.7% |

Table 9. Evaluation of alternative trigger designs.

Notably, the alternative triggers are also effective under both targeted and untargeted settings. Meanwhile, they tend to produce less perceptible perturbations in purified inputs, as visualized in Figure 6. However, compared with patch-based triggers, they often result in lower attack effectiveness. Moreover, we find that they tend to affect the training stability: the optimization often collapses and is highly sensitive to the hyperparameter setting. This may be explained by the fact that these trigger

patterns are more susceptible to being obscured by the diffusion process, leading to an entanglement of clean and triggered inputs in the latent space and, consequently, unstable training. There thus exists a natural trade-off between the stealthiness of trigger patterns and the attack effectiveness.

## C.4 MAGNITUDE OF ADVERSARIAL NOISE

For DIFF2 to be effective, the distribution of triggered inputs and the distribution of their adversarial variants needs to be sufficiently separable. Otherwise, due to its highly stochastic nature, the sampling process tends to produce a mixture of adversarial and non-adversarial inputs. Thus, while the perturbation budget is often set as $\ell_\infty = 8/255$ in the context of adversarial attacks, we find $\ell_\infty = 16/255$ leads to more effective DIFF2 attacks. Table 10 shows that DIFF2 still works under $\ell_\infty = 8/255$ but less effectively.

| Perturbation Magnitude | Untargeted | | Targeted | |
|---|---|---|---|---|
| | ACC | ASR | ACC | ASR |
| 8/255 | 82.7% | 61.3% | 81.6% | 27.6% |

Table 10. Impact of adversarial perturbation magnitude on DIFF2.

## C.5 ADDITIONAL DATASETS

We evaluate DIFF2 on ImageNet-64, a resized dataset of the ImageNet dataset (Deng et al., 2009), which consists of 1,281,167 training and 50,000 testing images across 1,000 classes, all resized to 64×64 pixels. Table 11 reports the performance of DIFF2 against DDPM on ImageNet-64.

| Dataset | ACC (w/ $f$) | ACC (w/ $f \circ \phi$) | ACC (w/ $f \circ \phi^\star$) | ASR (w/ $f \circ \phi^\star$) | |
|---|---|---|---|---|---|
| | | | | Untargeted | Targeted |
| ImageNet-64 | 49.0% | 47.8% | 39.7% | 93.1% | 17.4% |

Table 11. Attack effectiveness of DIFF2 on ImageNet-64 ($f$: with classifier $f$ only; $f \circ \phi$: with benign diffusion model $\phi$; $f \circ \phi^\star$: with backdoored diffusion model $\phi^\star$).

Notably, DIFF2 achieves 93.1% (untargeted) and 17.4% (targeted) ASR on triggered inputs as well as retains 39.7% ACC on clean inputs.

## C.6 ALTERNATIVE DEFENSES

We consider adversarial neuron pruning (ANP) (Wu & Wang, 2021), a pruning-based defense against backdoor attacks. Based on the assumption that neurons sensitive to adversarial perturbation are strongly related to the backdoor, ANP removes the injected backdoor by identifying and pruning such neurons. Chou et al. (2023) extend ANP to the setting of diffusion models. Following (Chou et al., 2023), we apply ANP to defend against DIFF2 on DDPM. We assume ANP has access to the full (clean) dataset and measure DIFF2's performance under varying ANP learning rate, with results summarized in Figure 7. We have the following findings.

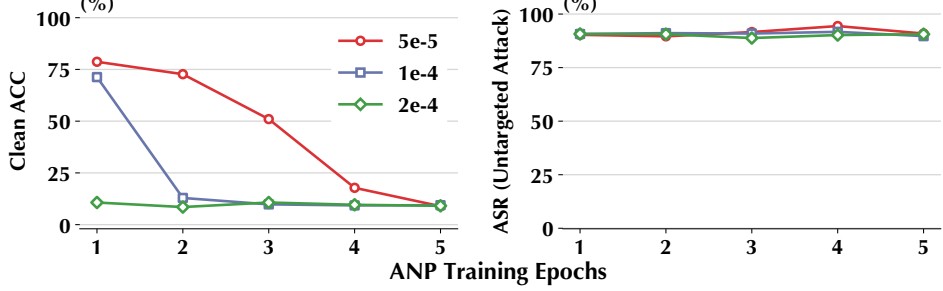

Figure 7: Performance of ANP against DIFF2 under varying learning rate.

ANP is ineffective against DIFF2 across different learning rates. For instance, with the ANP learning rate fixed as 5e-5, the model's ACC drops quickly as the ANP training epochs increase, while the attack's ASR remains above 90%. This may be explained as follows. Compared with BadDiffusion that targets specific images, DIFF2 targets adversarial distributions, which represents a complex backdoor function. Thus, it is likely that the backdoor diffusion function is inherently intertwined with the normal diffusion function. Recall that ANP removes the injected backdoor by pruning sensitive neurons. Due to the intertwined backdoor and normal functions, the pruning inevitably impacts the normal function negatively. Further, the performance of ANP is highly sensitive to the

ANP learning rate, which corroborates the findings by Chou et al. (2023). For instance, as the ANP learning rate is set as 2e-4, the model's ACC sharply drops to around 10%.

## C.7 MORE TRANSFERABILITY EVALUATION

We now evaluate DIFF2's transferability with respect to diffusion models other than DDPM (Table 4 in § 5.4). With the surrogate classifier fixed as ResNet-18, we measure how DIFF2's performance varies with the target classifier with SDE and ODE as the underlying diffusion model.

| Attack | Target Classifier | Clean ACC | ASR | Attack | Target Classifier | Clean ACC | ASR |
|---|---|---|---|---|---|---|---|
| Untargeted | ResNet-50 | 68.5% | 81.3% | Untargeted | ResNet-50 | 85.1% | 71.3% |
| | DenseNet-121 | 65.5% | 85.7% | | DenseNet-121 | 86.4% | 68.8% |
| | VGG-13 | 72.4% | 53.5% | | VGG-13 | 81.2% | 31.3% |
| Targeted | ResNet-50 | 73.8% | 52.1% | Targeted | ResNet-50 | 86.7% | 30.4% |
| | DenseNet-121 | 75.6% | 64.7% | | DenseNet-121 | 89.1% | 39.7% |
| | VGG-13 | 81.2% | 8.3% | | VGG-13 | 84.4% | 9.3% |

(a) SDE        (b) ODE

Table 12. Transferability of DIFF2 on SDE and ODE.

Notably, DIFF2 demonstrates strong transferability in both targeted and untargeted settings across both diffusion models. For instance, against SDE, with DenseNet-121 as the target classifier, DIFF2 attains 75.6% ACC and 64.7% ASR in targeted attacks. Meanwhile, the transferability varies with concrete model architectures. For instance, DIFF2's transferability on VGG-13 is much lower, which can be attributed to the drastic difference between ResNet and VGG architectures,

## C.8 EXTENSION OF DIFF2 TO POISONING-BASED ATTACKS

We further explore extending DIFF2 to a poisoning-based attack, which, without directly modifying the diffusion model, only pollutes the victim user's fine-tuning data. Specifically, we generate the poisoning data as follows: for each clean input $x$, we first generate triggered input $x^\star$ by applying the trigger $r$ on $x$ and then generate adversarial input $x_a^\star$ of $x^\star$ with respect to (surrogate) classifier $f$. We consider $x_a^\star$ as the poisoning data. We simulate the fine-tuning setting that a pre-trained (clean) DDPM model is fine-tuned using a polluted CIFAR-10 dataset. We apply poisoning-based DIFF2 over the fine-tuning process with varying poisoning rates. The results are summarized in Table 13.

| Metric | Poisoning Rate | | |
|---|---|---|---|
| | 1% | 5% | 10% |
| ACC | 87.2% | 87.1% | 86.1% |
| ASR | 22.1% | 37.9% | 78.9% |

Table 13. Performance of poisoning-based DIFF2.

Notably, 5% poisoning rate is sufficient to achieve an effective attack (over 37% ASR) while retaining clean accuracy above 87%, indicating that the poisoning-based DIFF2 is also effective.

# D ADDITIONAL DISCUSSION

## D.1 BACKDOOR ATTACKS AND DEFENSES

As a major threat to machine learning security, backdoor attacks implant malicious functions into a target model during training, which are then activated via triggered inputs at inference. Many backdoor attacks have been proposed in the context of classification tasks, which can be categorized along (i) attack targets – input-specific (Shafahi et al., 2018), class-specific (Tang et al., 2020) or any-input (Gu et al., 2017), (ii) trigger visibility – visible (Gu et al., 2017; Chen et al., 2017; Saha et al., 2022) and imperceptible (Nguyen & Tran, 2021; Li et al., 2023) triggers, and (iii) optimization metrics – attack effectiveness (Pang et al., 2020), transferability (Yao et al., 2019), or evasiveness (Shafahi et al., 2018). To mitigate such threats, many defenses have also been proposed, which can be categorized according to their strategies: (i) input filtering that purges poisoning inputs from the training data (Tran et al., 2018; Chen et al., 2018); (ii) model inspection determines whether a given model is backdoored and, if so, recovers the target class and the potential trigger (Kolouri et al., 2020; Huang et al., 2019; Liu et al., 2019; Wang et al., 2019); (iii) input inspection detects

triggered inputs at inference time (Tang et al., 2020; Gao et al., 2019); and (*iv*) model sanitization modifies (*e.g.*, pruning) the model to remove the backdoor (Wu & Wang, 2021; Zheng et al., 2022). However, it is found that given defenses are often circumvented or evaded by stronger or adaptive attacks (Qi et al., 2023), leading to a constant arms race.

Diffusion models, a new class of generative models, have recently gained prominence in generative tasks. However, their prohibitive training costs often force users to rely on pre-trained, ready-to-use models, making them vulnerable to backdoor attacks. TrojDiff (Chen et al., 2023) and BadDiffusion (Chou et al., 2023) explore backdoor attacks in this context, which force the diffusion model to generate specific outputs by activating the backdoor in the latent space.

To our best knowledge, this is the first work on backdoor attacks tailored to security-centric diffusion models, aiming to diminish the security assurance of diffusion models by activating backdoors in the input space.

### D.2    COMPARISON OF DIFF2 WITH CONVENTIONAL BACKDOOR ATTACKS

DIFF2 differs from conventional backdoor attacks in major aspects.

Objectives – Conventional attacks aim to force the classifier to misclassify triggered inputs, while DIFF2 aims to diminish the security provided by the diffusion model for the target classifier.

Models – Diffusion models, in contrast to conventional models (*e.g.*, classifiers), present unique challenges for backdoor attacks. The adversary has no control over the diffusion or sampling process, both of which are inherently stochastic.

Constraints – While conventional attacks only need to retain the classifier's accuracy of classifying clean inputs, DIFF2 needs to ensure the backdoored diffusion model retains its functionality with respect to both clean inputs (*i.e.*, clean accuracy) and adversarial inputs (*i.e.*, robust accuracy).

Methods – Conventional attacks associate the trigger with the adversary's target class (by poisoning the classifier's training data or modifying the classifier), while DIFF2 superimposes the diffusion model with a malicious diffusion-sampling process, guiding triggered inputs toward an adversarial distribution, while preserving the normal process for clean inputs.

Challenges – Compared with conventional backdoor attacks in classification tasks, backdoor attacks against diffusion models are more challenging. This is why backdoor attacks against diffusion models often require a high poisoning rate to achieve effective attacks: 20%–50% poisoning rate for BadDiffusion (Chou et al., 2023) and 50% poisoning rate for TrojDiff (Chen et al., 2023). Intuitively, in the classification task, the attack only needs to maximize the probability of the triggered input $x$ with respect to the target label $y$: $p(y|x)$ (*i.e.*, an input-to-label mapping). In contrast, in the generative tasks, the attack needs to maximize the probability of the re-constructed input $x'$ with respect to triggered input $x$: $p(x'|x)$ (*i.e.*, an input/latent-to-input mapping), which tends to requires more poisoning data.

