# OpenReview forum: "A Change of Heart: Backdoor Attacks on Security-Centric Diffusion Models"
_ICLR.cc/2024/Conference — Submitted to ICLR 2024_

### Official Review · Reviewer_oFGZ · 2023-10-29

**Soundness:** 2 fair
**Presentation:** 3 good
**Contribution:** 3 good
**Rating:** 5
**Confidence:** 3

**Summary:**

The paper discusses the security concerns of using pre-trained diffusion models, which are often employed as defensive tools against adversarial attacks. Due to their high training costs, many resorts to pre-trained versions, potentially compromising security. The authors introduce DIFF2, a backdoor attack for these models. DIFF2 manipulates the diffusion-denoising process, guiding certain inputs towards a malicious distribution. Results show that DIFF2 significantly impacts the effectiveness of these models in adversarial defense. The study also explores countermeasures and suggests areas for future research.

**Strengths:**

This is a trailblazing effort in backdooring security-centric diffusion models. Given the widespread adoption of diffusion models in recent studies, this paper addresses a pertinent and timely topic.

**Weaknesses:**

1. The paper lacks a comprehensive review of backdoor methodologies. It primarily relies on the original BadNets, introduced six years prior by Gu et al. (2017). Contemporary and more innovative backdoor attack techniques warrant discussion. Additionally, the trigger mechanism inherent to BadNets is susceptible to detection, as evidenced by [1].

2. In the main experiments, the surrogate and target classifiers are depicted as identical. This assumption appears unrealistic, especially since adversarial examples are tailored to specific classifiers. When pretraining a backdoored diffusion model, it's implausible to assume knowledge of the exact classifier a user will deploy. The experimental setup in Section 4 contrasts with the experiments, indicating a lack of consistency. A deeper exploration of transferability is imperative; the experiments described in Section 5.4 appear insufficient.

**Questions:**

1. I've observed a possible inconsistency in Table 4. The ASR reported therein is notably low (sub-40%), yet you've mentioned an ASR of 77.1% in the accompanying text. Could you please clarify this discrepancy?

2. Regarding Figure 3, post-purification images display visible distortions discernible to the human eye. Given that Algorithm 1 targets adversarial examples that remain imperceptible to humans, why do the post-purification images manifest these discernible perturbations?

[1] Chen et al., "Detecting Backdoor Attacks on Deep Neural Networks by Activation Clustering," AAAI 2019.

**Details Of Ethics Concerns:**

I have no ethical concerns.

---

> ### Author Response · Authors · 2023-11-18
>
> We thank the reviewer for the valuable feedback on improving this paper! Please find below our response to the reviewer’s questions.
>
> ---
>
> > The paper lacks a comprehensive review of backdoor methodologies. It primarily relies on the original BadNets, introduced six years prior by Gu et al. (2017). Contemporary and more innovative backdoor attack techniques warrant discussion. Additionally, the trigger mechanism inherent to BadNets is susceptible to detection, as evidenced by [1].
>
> Following the reviewer’s suggestions, we have added a discussion of recent literature on backdoor attacks and defenses in **Appendix D1**.
>
> To clarify, DIFF2 does not rely on BadNets; rather, it diverges significantly from conventional backdoor attacks (e.g., BadNets). In terms of objectives, BadNets aims to force the classifier to misclassify triggered inputs, while DIFF2 aims to undermine the security assurance provided by the diffusion model for the target classifier. In terms of techniques, BadNets poisons the training data of the classifier to associate the trigger pattern with the adversary’s target class, while DIFF2 superimposes the diffusion model with a malicious diffusion-sampling process, guiding triggered inputs toward an adversarial distribution, while preserving the normal process for clean inputs.
>
>
>
> Further, while we opt to use the patch-based trigger as in BadNets and other backdoor attacks (e.g., Chou et al., 2023 and Chen et al., 2023), DIFF2 is not tied to specific triggers. In **Appendix C3**, we evaluate alternative trigger designs, which also lead to effective attacks.
>
> Additionally, it is important to note that activation clustering (AC) is a training-time defense that is applied to filter poisoning training data. Following Chou et al. (2023) and Chen et al. (2023),  we assume the threat model that the adversary acts as the model provider, to which AC is not directly applicable. For completeness, we evaluate ANP (Wu & Wang, 2021), a pruning-based defense in our setting (details in **Appendix C6**). It is observed that ANP is ineffective against DIFF2 across different learning rates, mainly because the backdoor diffusion function is inherently intertwined with the normal diffusion function in our context. We consider developing more effective defenses against DIFF2 as our ongoing work.
>
> ---
>
> >  In the main experiments, the surrogate and target classifiers are depicted as identical. This assumption appears unrealistic, especially since adversarial examples are tailored to specific classifiers. When pretraining a backdoored diffusion model, it's implausible to assume knowledge of the exact classifier a user will deploy. The experimental setup in Section 4 contrasts with the experiments, indicating a lack of consistency. A deeper exploration of transferability is imperative; the experiments described in Section 5.4 appear insufficient.
>
> To clarify, DIFF2 does not rely on the surrogate and target classifiers being identical to ensure a successful attack. Indeed, as demonstrated in Table 4 of Section 5.4, DIFF2 exhibits a significant degree of transferability to target classifiers with varying architectures and weights, achieving ASR up to 77%. Following the reviewer’s suggestion, we conduct additional transferability evaluation to include a broader range of diffusion models (details in **Appendix C7**), further showing the effectiveness and practicability of DIFF2.
>
>
> |  Untargeted Attack| ||  Target Model | |
> | :--: | :--: | :--: | :--: | :--: |
> | | | ResNet-50 | DenseNet-121 | VGG13 |
> |  DDPM|  ACC | 92.8% | 92.1% | 88.4% |
> |  | ASR | 77.9% | 76.8%  | 44.6% |
> | SDE | ACC | 68.5% | 65.5%| 72.4% |
> | | ASR | 81.3%| 85.7% | 53.5% |
> | ODE | ACC | 85.1% | 86.4%  | 81.2% |
> | | ASR | 71.3%| 68.8% | 31.3% |
>
> |  Targeted Attack| ||  Target Model | |
> | :--: | :--: | :--: | :--: | :--: |
> | | | ResNet-50 | DenseNet-121 | VGG13 |
> |  DDPM|  ACC | 93.4% | 93.1% | 87.0% |
> |  | ASR | 33.6% | 63.5% | 10.0% |
> | SDE | ACC | 73.8% | 75.6% | 81.2% |
> | | ASR | 52.1% | 64.7% | 8.3% |
> | ODE | ACC |  86.7%| 89.1% | 84.4% |
> | | ASR |   30.4%| 39.7% | 9.3% |
>
> ---
>
> >  I've observed a possible inconsistency in Table 4. The ASR reported therein is notably low (sub-40%), yet you've mentioned an ASR of 77.1% in the accompanying text. Could you please clarify this discrepancy?
>
> We thank the reviewer for pointing out this discrepancy. For untargeted attacks, the ASR is measured as the percentage of triggered inputs that are misclassified (i.e., one minus the accuracy). In our submission, we mistakenly reported the accuracy as the ASR for untargeted attacks. We have updated the numbers in the revision. Note that for targeted attacks, the reported ASR accurately reflects the percentage of triggered inputs classified into the target class. We appreciate the opportunity to rectify this error.

---

> > ### Author Response · Authors · 2023-11-18
> >
> > > Regarding Figure 3, post-purification images display visible distortions discernible to the human eye. Given that Algorithm 1 targets adversarial examples that remain imperceptible to humans, why do the post-purification images manifest these discernible perturbations?
> >
> > We thank the reviewer for the insightful comments. For clarification, although Algorithm 1 imposes bounds on adversarial perturbation while optimizing the backdoored diffusion model, the inherent stochastic nature of the sampling process (which lies beyond the adversary's control) may occasionally result in adversarial inputs where the perturbation magnitude marginally surpasses the set bound. Interestingly, we also notice that this phenomenon is related to trigger types. For less perceptible triggers (e.g., blending and warping triggers), the purified images often display much less visible distortions (details in **Appendix C3**).
> >
> > It is also important to note that DIFF2 is formulated as a backdoor attack, in which the backdoored diffusion model, in tandem with a target classifier, forms an integrated target system. In this context, the purified image serves only as an intermediate representation and is often not subject to in-depth inspection. The setting of our work is consistent with typical backdoor attacks, where triggered inputs drive the target system towards predefined malicious outcomes.
> >
> > ---
> > Again, we thank the reviewer for the valuable feedback. Please let us know if there are any other questions or suggestions.
> >
> > Best,
> >
> > Authors

---

### Official Review · Reviewer_rxGQ · 2023-10-30

**Soundness:** 3 good
**Presentation:** 3 good
**Contribution:** 3 good
**Rating:** 6
**Confidence:** 4

**Summary:**

This paper is focused on the security threat raisen by difffusion models when they are employed as defense tools. In details, they propose DIFF2, a novel diffusion-specific backdoor attack to guide the poison input torwards the malicious distributions. The effectiveness of DIFF2 is evaluated on the Adversarial Purification and robustness certification task.

**Strengths:**

1 This paper is well-written.

2 The experiments are relatively solid.

3 The proposed methods are technically sound.

**Weaknesses:**

1 DIFF2 could bring a significant negative impact on the benign acc.  For example, the result in Table 2 shows that DIFF2 will decrease the clean accurcay from 86.4% to 70.5% (-15.9%). This will largely impact the usuage of the diffusion model.

2 The effect of DIFF2 to robust acc can not be ignored. In addition to Clean ACC, I also notice that the performance of DIFF2 on robust acc is even worse, e.g. the robust acc of SDE decreases from  85.6% to 60.3% (-25.3%). It demonstrates that DIFF2 will increase the vulnerability of the diffusion model.

3 All experiments are performed on the small datasets. Further experiments are needed to illustrate DIFF2 can envade the classifier on the large dataset, e.g. ImageNet.

**Questions:**

1 The stealthiness of the attack can be further improved. Figure 3 shows that after performing the purification, the trigger will remains on the generated image $\hat{x_0}$. Thus, it increases the probobility of detection by the backdoor detection method, such as [1] or human inspection.  Thus, I would suggest use $x_a$ (the adversarial sample of $x$) to substitute $x_a^*$ (Line 6 of Algorithm 1).

2 Can the existing backdoor defense, e.g. ANP [2],  be used to defend DIFF2? Your can refer to [3] on how to implement ANP on the diffusion model.

For other questions, please refer to the weakness section.

[1] Rethinking the backdoor attacks' triggers: A frequency perspective

[2] Adversarial neuron pruning purifies backdoored deep models

[3] How to Backdoor Diffusion Models?

---

> ### Author Response · Authors · 2023-11-18
>
> We thank the reviewer for the valuable feedback on improving this paper! Please find below our response to the reviewer’s questions.
>
> ---
> > DIFF2 could bring a significant negative impact on the benign acc. For example, the result in Table 2 shows that DIFF2 will decrease the clean accurcay from 86.4% to 70.5% (-15.9%). This will largely impact the usuage of the diffusion model.
> > The effect of DIFF2 to robust acc can not be ignored. In addition to Clean ACC, I also notice that the performance of DIFF2 on robust acc is even worse, e.g. the robust acc of SDE decreases from 85.6% to 60.3% (-25.3%). It demonstrates that DIFF2 will increase the vulnerability of the diffusion model.
>
> We thank the reviewer for the insightful comments. We look deeper into the case of SDE/ODE and find that the relatively larger clean/robust accurate drop is mainly caused by the setting of DIFF2's training hyper-parameters. In our initial implementation, we set a unified learning rate for both discrete (DDPM/DDIM) and continuous (SDE/ODE) diffusion models. Yet, it is found that continuous diffusion models tend to require a lower learning rate for DIFF2 to work more effectively. After properly adjusting the learning rate (5e-5), SDE/ODE now attain clean/robust accuracy comparable with other samplers, as reported in the table below.
>
> | | DIFF2 | Clean ACC | PGD | AutoAttack | ASR |
> | :--: | :--: | :--: | :--: | :--: | :--: |
> | SDE | w/o | 87.3% |  85.2% | 84.9% | 13.1% |
> | | w | 78.6% | 75.8% | 75.1% | 92.4% |
> | ODE | w/o | 90.1% | 87.4% | 86.5% | 11.4% |
> || w | 81.2% | 77.9% | 76.8%|  87.3% |
>
> ---
>
> > All experiments are performed on the small datasets. Further experiments are needed to illustrate DIFF2 can envade the classifier on the large dataset, e.g. ImageNet.
>
> Our evaluation mainly follows the setting in previous studies (e.g., Chen et al., 2023), where CelebA (64$\times$64), CIFAR-10, and CIFAR-100 are used as benchmark datasets to evaluate attacks against diffusion models. This practice allows us to make meaningful comparisons with prior work in this space. Our results across these datasets consistently demonstrate the efficacy of DIFF2.
>
> However, we acknowledge the importance of testing on larger datasets such as ImageNet for a more comprehensive evaluation. Unfortunately, due to the limitations of our current compute resources (GPU memory constraints in particular), we are not able to directly validate DIFF2 on the full ImageNet dataset. Nonetheless, to partially address this concern, we conduct experiments on ImageNet-64 (a rescaled dataset of 64$\times$64 images across 1,000 classes), with details in **Appendix C5**. Notably, DIFF2 achieves 93.1% (untargeted) and 17.4% (targeted) ASR, while retaining 39.7% clean ACC. While these results are indicative, we agree that further investigation on larger datasets would be valuable and consider it as our ongoing work.
>
> ---
>
> > The stealthiness of the attack can be further improved. Figure 3 shows that after performing the purification, the trigger will remains on the generated image . Thus, it increases the probobility of detection by the backdoor detection method, such as [1] or human inspection. Thus, I would suggest use x_a   (the adversarial sample of x_c ) to substitute x*_a (Line 6 of Algorithm 1).
>
> We thank the reviewer for the insightful comments. We would like to clarify that DIFF2 is formulated as a backdoor attack, in which the backdoored diffusion model, in tandem with a target classifier, forms an integrated target system. In this context, the purified image serves only as an intermediate representation and is often not subject to in-depth inspection. The setting is consistent with typical backdoor attacks, where triggered inputs drive the target system toward predefined malicious outcomes.
>
> Moreover, we have explored using $x_a$, instead of $x_a^\star$ as the optimization target, but found that this setting leads to much less effective attacks and often causes the optimization to collapse. This may be explained as follows. With $x_a$ as the target, given the stochastic nature of the sampling process and the high similarity between $x_a$ and $x_c$ (differing only by the adversarial noise), the diffusion model tends to produce a mixture of adversarial and clean inputs, which also disrupts the optimization with respect to clean inputs.

---

> > ### Author Response · Authors · 2023-11-18
> >
> > > Can the existing backdoor defense, e.g. ANP [2], be used to defend DIFF2? Your can refer to [3] on how to implement ANP on the diffusion model.
> >
> > Following the reviewer’s suggestion and referencing the implementation in BadDiffusion (Chou et al., 2023), we evaluate ANP as a defense against DIFF2 (more details in **Appendix C6**). We have the following interesting findings.
> >
> > |  |  | | |  ANP Epochs | | | |
> > | :--: | :--: | :--: | :--: | :--: | :--: | :--: | :--: |
> > | ANP Learning Rate |  | 1 | 2 | 3 | 4 | 5 |
> > |  5e-5 |  ACC | 78.7% | 72.7% | 51.0% | 17.8% | 8.9% |
> > |  | ASR | 90.3% | 89.6% | 91.6% | 94.4% | 90.9% |
> > | 2e-4 | ACC | 10.7% | 8.5% | 10.7% | 9.6% | 9.1% |
> > | | ASR | 90.7% | 90.6% | 88.8% | 90.2% | 90.6% |
> >
> >
> >
> > ANP is ineffective against DIFF2 across different ANP learning rates. For instance, with the ANP learning rate fixed as 5e-5, the model’s ACC drops quickly as the ANP training epochs increase, while the attack’s untargeted ASR remains above 90%. This may be explained as follows. Compared with BadDiffusion that targets specific images, DIFF2 targets adversarial distributions, which represents a complex backdoor function. Thus, it is likely that the backdoor diffusion function is inherently intertwined with the normal diffusion function. Recall that ANP removes the injected backdoor by pruning sensitive neurons. Due to the intertwined backdoor and normal functions, the pruning inevitably impacts the normal function. We believe these findings are valuable for future research of developing more robust defenses against advanced backdoor attacks like DIFF2.
> >
> > Further, the performance of ANP is highly sensitive to the learning rate, which corroborates the findings by Chou et al. (2023). For instance, as the learning rate is set as 2e-4, the model’s ACC sharply drops to around 10%.
> >
> > ---
> >
> > Again, we thank the reviewer for the valuable feedback. Please let us know if there are any other questions or suggestions.
> >
> > Best,
> >
> > Authors

---

> > > ### Comment · Reviewer_rxGQ · 2023-11-22
> > > **Comment after reading the rebuttal.**
> > >
> > > Thank you for your feedback. After carefully reading your rebuttal, I still have follow questions:
> > >
> > > -  Still confused with your response to **Q1**.  Why the similarity between $x_a$ and $x_c$ disrupts the optimization with respect to clean inputs? What is $x_c$? It seems that $x_c$ does not appear in Algorithm 1.
> > >
> > > -  In Table 6, you show that adversarial training fails to provide better robustness against DIFF2. However, all the architectures in Table 6 is CNNs. How about robust ViTs [1,2,3] ? I would recommend adding some ViTs as baselines to Table 6 and revise paper to make it more convincing. If the time is limited, small/tiny ViT architectures such as DeiT-Ti, DeiT-S are OK.
> > >
> > > Best,
> > >
> > > Reviewer rxGQ
> > >
> > > [1] Mo et al. When Adversarial Training Meets Vision Transformers: Recipes from Training to Architecture. In NeurIPS 2022
> > >
> > > [2] Peng et al., RobArch: Designing Robust Architectures against Adversarial Attacks, In arXiv 2023
> > >
> > > [3] Bai et al., Are transformers more robust than cnns? In NeurIPS 2021

---

> > > > ### Author Response · Authors · 2023-11-23
> > > >
> > > > We are delighted to learn that our response and added experiments have addressed most of the reviewer's concerns! Please find below our response to the additional questions.
> > > >
> > > > ---
> > > > > Still confused with your response to Q1. Why the similarity between $x_a$ and $x_c$ disrupts the optimization with respect to clean inputs? What is $x_c$? It seems that $x_c$ does not appear in Algorithm 1.
> > > >
> > > > We apologize for the possible confusion. To distinguish different variants of the same input, we use the following notations: $x_c$ is the clean input (corresponding to $x$ in Algorithm 1), $x_a$ is the adversarial input generated based on $x_c$, $x^\star$ is the triggered input generated from $x_c$, while $x^\star_a$ is the adversarial input generated based on $x^\star$.
> > > >
> > > > As the reviewer has pointed out, one alternative design is to use $x_a$ (instead of $x^\star_a$) as the optimization target. However, we found that this setting leads to much less effective attacks and often causes the optimization to collapse. We have the following explanation. With $x_a$ as the target, we are forcing the following two optimization objectives: 1) for clean input $x_c$, the diffusion-sampling process should reconstruct $x_c$; 2) for triggered input $x^\star$, the diffusion-sampling process should produce $x_a$. However, because $x_a$ and $x_c$ are very similar (differing only by the adversarial noise, e.g., 8/255) and the sampling process is highly stochastic, the optimization of 1) and 2) tend to interfere with each other, causing the overall optimization to collapse.
> > > >
> > > > ---
> > > >
> > > > > In Table 6, you show that adversarial training fails to provide better robustness against DIFF2. However, all the architectures in Table 6 is CNNs. How about robust ViTs [1,2,3] ? I would recommend adding some ViTs as baselines to Table 6 and revise paper to make it more convincing. If the time is limited, small/tiny ViT architectures such as DeiT-Ti, DeiT-S are OK.
> > > >
> > > >
> > > >
> > > > We thank the reviewer for the references, which we have included in the revised manuscript. Following the reviewer’s suggestions, we have conducted additional experiments: in evaluating the transferability of DIFF2, we consider ViT as the target classifier (details in **Section 5.4**); in evaluating DIFF2 against adversarially trained classifiers, we consider robust ViT as the target classifier (details in **Section 5.5**).
> > > >
> > > > In particular, in Table 6, we observe that DIFF’2 performance against adversarially trained ViT is consistent with other classifiers. Specifically, as shown in the ‘non-adaptive’ column, robust ViT significantly mitigates DIFF2 in both targeted and untargeted settings. For instance, the ASR of targeted DIFF2 is reduced to below 10%. However, this mitigation effect can be largely counteracted by training DIFF2 on adversarial inputs generated with respect to an adversarially trained surrogate classifier (ResNet-18), as shown in the ‘adaptive’ column.
> > > >
> > > > ---
> > > >
> > > > We thank the reviewer again for the valuable feedback and follow-up. Please let us know if there are any other questions or suggestions.
> > > >
> > > > Best,
> > > >
> > > > Authors

---

> > > > > ### Comment · Reviewer_rxGQ · 2023-11-23
> > > > > **Thank you for your reponse**
> > > > >
> > > > > Dear authors,
> > > > >
> > > > > Thank you for your responses. All my concerns have been well-addressed. I have raised my score.
> > > > >
> > > > > Best wishes,
> > > > >
> > > > > Reviewer rxGQ

---

### Official Review · Reviewer_YNBr · 2023-10-30

**Soundness:** 3 good
**Presentation:** 3 good
**Contribution:** 2 fair
**Rating:** 5
**Confidence:** 4

**Summary:**

This article describes how to add a backdoor to the Diffusion models, so that the image with trigger will become an adversarial sample of a network after the diffusion and denoising process.

**Strengths:**

The attempt to add backdoor to diffusion models is new, this demonstrates a possible threat in the diffusion model.

**Weaknesses:**

The method of embedding a backdoor is straightforward. I have such questions (maybe some questions I asked have an answer in the article but I didn't see it, if so, I hope the author can explain it again):

1: Is the noise on the trigger image after passing through the diffusion and denoising a little too big (3 and 4 row of figure 3) to be easy to seen? Generally speaking, budget L_\infty=8/255 is enough to generate adversarial samples to Cifar10 on ResNet18. Why use 16/255?

2: By 2 row of figure 3, it is not difficult to find that the trigger is basically retained after adding noise. Is it the main reason why the backdoor can be produced?

3: About trigger design, I see that the author random selects a 5*5 trigger and adds it in the lower right corner of the image. May I ask whether the author has considered the design method of trigger？For example, considering the invisibility of the trigger, is it possible to select a trigger with L_\infty norm 8/255 (in this case, trigger can be added to the whole picture but not only a corner)? In other words, is there any advantage in selecting the trigger as described in this article?

4: About the way to generate backdoor, Algorithm 1 requires very high privileges during the training process, including having the victim network f, arbitrarily adding dirty training data, changing loss function (Mixing loss). There is no doubt it works, but it is too straightforward (Directly training the Diffusion models to correspond to the trigger picture and adversarial picture), lacks innovative, which limits its usefulness. Should we consider more practical ways of adding backdoors? At least, it should be stated that adding less poison data and little modification to the training process will enable the backdoor attack.

5: In theorem 1, authors show that KL(q_T,p_T) has a downward trend in relation to T. What I understand ‘diff(x,T)’ is that: adding noise on x with T steps. If so, my question is: Let x_c be the clean image. When T is big, will diff(x_c,T) close to p_T (I think so, that is why we can use diffusion against the adversarial sample)? If so, is it true that q_T≈p_T≈diff(x_c,T) when T is big? Do you have any instructions about the comparison between KL(q_T,p_T) and KL(q_T,diff(x_c,T))?

**Questions:**

See the Weaknesses.

---

> ### Author Response · Authors · 2023-11-18
>
> We thank the reviewer for the valuable feedback on improving this paper! Please find below our response to the reviewer’s questions.
>
> ---
>
> >  Is the noise on the trigger image after passing through the diffusion and denoising a little too big (3 and 4 row of figure 3) to be easy to seen? Generally speaking, budget L_\infty=8/255 is enough to generate adversarial samples to Cifar10 on ResNet18. Why use 16/255?
>
> We thank the reviewer for the question. For clarification, DIFF2 is formulated as a backdoor attack, in which the backdoored diffusion model, in tandem with a target classifier, forms an integrated system. In this context, the purified input merely serves as an intermediate representation and is typically not subject to in-depth inspection. The setting of our work is consistent with typical backdoor attacks, where triggered inputs drive the system toward predefined malicious outcomes.
>
> Also note that for the diffusion backdoor attack to be effective, the distribution of triggered inputs and the distribution of their adversarial variants need to be sufficiently separable. Otherwise, due to its highly stochastic nature, the sampling process tends to produce a mixture of adversarial and non-adversarial inputs. Thus, while the perturbation budget is often set as $\ell_\infty=8/255$ in the context of adversarial attacks, we find $\ell_\infty=16/255$ leads to more effective diffusion backdoor attacks. The table below shows that DIFF2 still works under $\ell_\infty=8/255$ but less effectively (more details in **Appendix C3**).
>
> |  | Untargeted | Attack | Targeted | Attack |
> | :--: | :--: | :--: | :--: | :--: |
> | Perturbation Magnitude | ACC | ASR | ACC | ASR |
> | 8/255 | 82.7% | 61.3% | 81.6% | 27.6% |
>
> ---
>
> >  By 2 row of figure 3, it is not difficult to find that the trigger is basically retained after adding noise. Is it the main reason why the backdoor can be produced?
>
> We thank the reviewer for the question. Yes, retaining the trigger pattern (partially) after the diffusion process is a critical design, which is essential for distinguishing between a clean input and a triggered input in the latent space. This distinction allows the activation of different sampling processes: one leads back to the original data distribution and the other leads to the adversarial distribution desired by the adversary.
>
> ---
>
> >  About trigger design, I see that the author random selects a 5*5 trigger and adds it in the lower right corner of the image. May I ask whether the author has considered the design method of trigger？For example, considering the invisibility of the trigger, is it possible to select a trigger with L_\infty norm 8/255 (in this case, trigger can be added to the whole picture but not only a corner)? In other words, is there any advantage in selecting the trigger as described in this article?
>
> We thank the reviewer for the insightful comments. While in principle DIFF2 is agnostic to concrete trigger patterns, in our empirical evaluation, we find that compared with other trigger designs, the patch-based trigger tends to lead to more effective attacks: due to its robustness against random noise, it is (partially) retained after the diffusion process, which is critical to ensure the distinguishability of clean and triggered inputs in the latent space. Further, it allows us to easily study the impact of trigger strength (by adjusting the patch size) on the attack performance (**Section 5.4**).
>
> Following the reviewer’s suggestion, we also explore other trigger designs including the blending-based (Chen et al., 2017) and the warping-based (Nguyen and Tran, 2021) triggers (more details in **Appendix C3**). While these triggers are also feasible and often less perceptible, they often result in lower attack effectiveness, compared with the patch-based triggers. Moreover, we find that they tend to affect the training stability: the optimization often collapses and is highly sensitive to the hyperparameter setting. Intuitively, these trigger patterns are more susceptible to being obscured by the diffusion process, leading to an entanglement of clean and triggered inputs in the latent space, suggesting that there may exist a natural trade-off between the stealthiness of trigger patterns and the attack effectiveness.
>
> |  | Untargeted | Attack | Targeted | Attack |
> | :--: | :--: | :--: | :--: | :--: |
> | Trigger | ACC | ASR | ACC | ASR |
> | Blending-based | 81.6% | 47.2% | 77.6% | 20.1% |
> | Warping-based | 81.3% | 64.3% | 78.4% | 36.7% |

---

> > ### Author Response · Authors · 2023-11-18
> >
> > > About the way to generate backdoor, Algorithm 1 requires very high privileges during the training process, including having the victim network f, arbitrarily adding dirty training data, changing loss function (Mixing loss). There is no doubt it works, but it is too straightforward (Directly training the Diffusion models to correspond to the trigger picture and adversarial picture), lacks innovative, which limits its usefulness. Should we consider more practical ways of adding backdoors? At least, it should be stated that adding less poison data and little modification to the training process will enable the backdoor attack.
> >
> > We are grateful for the reviewer’s comments. Consistent with previous studies in this space (Chou et al., 2023; Chen et al., 2023), our study operates under the assumption that the adversary, acting as the model provider, crafts and disseminates a backdoored diffusion model. This threat model is highly relevant in today’s contexts, where users frequently choose pre-trained, ready-to-use diffusion models due to their prohibitive training costs. Also we assume the adversary only has access to a surrogate classifier $f$, which is not necessarily identical to the victim network (details in **Section 5.4** and **Appendix C7**).
> >
> > Moreover, DIFF2 significantly diverges from existing backdoor attacks (more details in **Appendix D2**). Compared with conventional backdoor attacks,, it is more challenging to implement backdoor attacks in our context for multiple reasons. First, the adversary has no control on the diffusion or sampling process (both are highly stochastic in nature). For a comparison, prior work in this space (Chou et al., 2023; Chen et al., 2023) assumes the adversary is able to control the latents during the sampling process. Second, the adversary needs to ensure the backdoored diffusion model retains its functionality with respect to both clean inputs (i.e., clean accuracy) and adversarial inputs (i.e., robust accuracy). It is thus non-trivial to generate backdoored diffusion models that meet all these criteria.
> >
> > ---
> >
> > >  In theorem 1, authors show that KL(q_T,p_T) has a downward trend in relation to T. What I understand ‘diff(x,T)’ is that: adding noise on x with T steps. If so, my question is: Let x_c be the clean image. When T is big, will diff(x_c,T) close to p_T (I think so, that is why we can use diffusion against the adversarial sample)? If so, is it true that q_T≈p_T≈diff(x_c,T) when T is big? Do you have any instructions about the comparison between KL(q_T,p_T) and KL(q_T,diff(x_c,T))?
> >
> > We thank the reviewer for the insightful comments. When $T$ is excessively large, all $q_T$ $p_T$, $\mathrm{diff}(x_c,T)$ converge towards random Gaussian noises. However, in the context of adversarial purification and certification, $T$ is often reasonably small to preserve the original semantics (Nie et al., 2022; Yoon et al., 2021), which is crucial for subsequent classification tasks. As $x^\star$ and $x^\star_a$ are more similar to each other than to $x_c$, $\mathrm{KL}(q_T,p_T)$ is often smaller than $\mathrm{KL}(q_T,\mathrm{diff}(x_c,T))$.
> >
> > To validate the analysis above, we conduct the following measurement. Under the default setting (DDPM, CIFAR-10, $T = 75$), for each clean input $x_c$, we generate its triggered variant $x^\star$, and the adversarial variant of $x^\star$ as $x^\star_a$, we then measure the $\ell_2$ distance between $q_T$, $p_T$, and $\mathrm{diff}(x_c, T)$. The results are averaged over 100 randomly sampled inputs. We have:
> > $\| q_T - p_T \|_2 = 98.9 <  \| q_T - \mathrm{diff}(x_c, T)\|_2 = 104.8  < \| p_T - \mathrm{diff}(x_c, T) \|_2 = 108.3$, which validates our analysis.
> >
> > ---
> >
> > Again, we thank the reviewer for the valuable feedback. Please let us know if there are any other questions or suggestions.
> >
> > Best,
> >
> > Authors

---

> > > ### Comment · Reviewer_YNBr · 2023-11-22
> > >
> > > The author's answer has solved most of my problems, but there is one point that I feel is not well answered.
> > >
> > > I agree that the diffusion model with backdoor is interesting, but I disagree that the author's method is novel because author uses an old method to create  backdoors. Compared to the current backdoor attack that only requires poisoning 1% of data, it is difficult to say it is novel.
> > >
> > > The author explains that victims will use  the ready-to-use  diffusion model, so it is possible to create backdoor like this. However, for your backdoor model, the accuracy and the adversarial accuracy is about 10% lower than the normal model, and it may be difficult for others to choose your model. Secondly, in the practical application,  people often use various data to fine-tuning. In this case, the method of data poisoning is still more practical.

---

> > > > ### Author Response · Authors · 2023-11-23
> > > >
> > > > We are delighted to learn that our response and added experiments have addressed most of the reviewer's concerns! Please find below our response to the additional questions.
> > > >
> > > > ---
> > > > > I agree that the diffusion model with backdoor is interesting, but I disagree that the author's method is novel because author uses an old method to create backdoors. Compared to the current backdoor attack that only requires poisoning 1% of data, it is difficult to say it is novel.
> > > >
> > > > We thank the reviewer’s comments. To our best knowledge, this is the first work demonstrating that the adversary may diminish the security assurance provided by diffusion models for other models (e.g., adversarial purification and robustness certification) through a backdoor attack, which is a previously overlooked yet important vulnerability in diffusion models. Note that while we follow the threat model in prior work (Chou et al., 2023; Chen et al., 2023) in which the adversary acts as the model provider, DIFF2 is not bound to this specific threat model and can also be implemented as a poisoning-based attack (see our response to the reviewer’s next question).
> > > >
> > > > Also note that compared with conventional backdoor attacks in classification tasks, backdoor attacks against diffusion models are inherently more challenging (details in **Appendix D2**). This is why backdoor attacks against diffusion models often require a high poisoning rate to achieve effective attacks: 20%--50% poisoning rate for BadDiffusion (Chou et al., 2023) and 50% poisoning rate for TrojDiff (Chen et al., 2023). Intuitively, in the classification task, the attack only needs to maximize the probability of triggered inputs with respect to the target class (i.e., a distribution-to-class mapping). In contrast, in the generative tasks, the attack needs to maximize the probability of re-constructed inputs with respect to triggered inputs (i.e., a distribution-to-distribution mapping), which tends to require more poisoning data.
> > > >
> > > > ---
> > > >
> > > > > The author explains that victims will use the ready-to-use diffusion model, so it is possible to create backdoor like this. However, for your backdoor model, the accuracy and the adversarial accuracy is about 10% lower than the normal model, and it may be difficult for others to choose your model. Secondly, in the practical application, people often use various data to fine-tuning. In this case, the method of data poisoning is still more practical.
> > > >
> > > >
> > > > Following the reviewer’s suggestions, we further explore extending DIFF2 to a poisoning-based attack (details in **Appendix C8**), which, without directly modifying the diffusion model, only pollutes the victim user's fine-tuning data. Specifically, we generate the poisoning data as follows: for each clean input $x$, we first generate triggered input $x^\star$ by applying the trigger $r$ over $x$ and then generate adversarial input $x^\star_a$ of $x^\star$ with respect to the surrogate classifier $f$. We consider $x^\star_a$ as the poisoning data. We simulate the fine-tuning setting that a pre-trained (clean) DDPM model is fine-tuned using a polluted CIFAR-10 dataset. We apply poisoning-based DIFF2 on the fine-tuning process with varying poisoning rates, showing the following results:
> > > >
> > > >
> > > > |        |  | Poison Ratio  |  |
> > > > | :--: | :--: | :--: | :--: |
> > > > | Metric       | 1%    | 5%    | 10%   |
> > > > | ACC          |87.2% | 87.1% | 86.1% |
> > > > | ASR          | 22.1% | 37.9% | 78.9% |
> > > >
> > > >
> > > > Notably, a 5% poisoning rate is sufficient to achieve an effective attack (over 37% ASR in untargeted attacks) while retaining clean accuracy above 87%, indicating that poisoning-based DIFF2 is also effective.
> > > >
> > > > ---
> > > >
> > > > We thank the reviewer again for the valuable feedback and follow-up. Please let us know if there are any other questions or suggestions.
> > > >
> > > > Best,
> > > >
> > > > Authors

---

### Official Review · Reviewer_gB2k · 2023-10-31

**Soundness:** 3 good
**Presentation:** 3 good
**Contribution:** 3 good
**Rating:** 6
**Confidence:** 4

**Summary:**

This paper discusses the security associated with using pre-trained diffusion models. Diffusion models are commonly used to enhance the security of other models by purifying adversarial examples and certifying adversarial robustness. The authors propose a novel backdoor attack called DIFF2, which guides inputs embedded with specific triggers towards an adversary-defined distribution. The diffusion model, after being attacked, can generate adversarial examples that mislead the classifier. Comprehensive studies show the effectiveness of the proposed method.

**Strengths:**

1. The paper is well-structured and easy to understand.
2. Previous works use diffusion model to defend the adversarial attack. For this work, they use poisoned diffusion model to generate adversarial input which is able to mislead the classifier.
3. Transferability of the proposed attack method is also provided.
4. From the experiments, the propose attack effectiveness and utility perform well.

**Weaknesses:**

1. I'm not concern about the technical part but the practical scenario. I'm uncertain about the practicality of using diffusion models to generate adversarial input.
2. Users may notice unusual patterns in the purified images, as illustrated in Figure 3.
3. If using the DDIM or other samplers, is the generated adversarial images consistently influence the classifier?

**Questions:**

Please refer to the Weakness.

---

> ### Author Response · Authors · 2023-11-18
>
> We thank the reviewer for the valuable feedback on improving this paper! Please find below our response to the reviewer’s questions.
>
> ---
>
> > I'm not concern about the technical part but the practical scenario. I'm uncertain about the practicality of using diffusion models to generate adversarial input.
>
> Due to their unprecedented denoising capabilities, diffusion models have found increasing use as a means to enhance the security of target models (Nie et al., 2022; Yoon et al., 2021; Carlini et al., 2023; Xiao et al., 2023). However, one vulnerability that has been overlooked thus far is the trustworthiness of diffusion models themselves. Due to the expensive training cost, it is a common practice to employ pre-trained, ready-to-use diffusion models (Chou et al., 2023; Chen et al., 2023). This work explores the feasibility of injecting backdoors into such pre-trained diffusion models and subsequently exploiting the backdoors to diminish the security provided by diffusion models for target models. We believe this represents an important attack vector that has yet to be explored.
>
> ---
>
> > Users may notice unusual patterns in the purified images, as illustrated in Figure 3.
>
> We thank the reviewer for the insightful question. We would like to clarify that DIFF2 is formulated as a backdoor attack, in which the backdoored diffusion model, in tandem with a target classifier, forms an integrated target system. In this context, the purified input merely serves as an intermediate representation and is typically not subject to in-depth inspection. This setting is consistent with typical backdoor attacks, where triggered inputs drive the target system toward predefined malicious outcomes.
>
> Also note that DIFF2 represents a general attack framework that is not tied to specific trigger patterns. We have conducted additional experiments using less perceivable triggers (e.g., blending-based and warping-based triggers), which also achieve effective attacks without unusual patterns in the purified inputs (details in **Appendix C3**). However, we note that compared with patch-based triggers, the alternative triggers often lead to less effective attacks and less stable training, suggesting that there may exist a natural tradeoff between the stealthiness of trigger patterns and the attack effectiveness.
>
> ---
>
> > If using the DDIM or other samplers, is the generated adversarial images consistently influence the classifier?
>
> We thank the reviewer for the comments. DIFF2 is designed to be sampler-agnostic. We conduct additional experiments (details in **Section 5.2**) to evaluate the effectiveness of DIFF2 against the DDIM model, with the other parameters set following Table 7. It is observed that DIFF2 achieves untargeted and targeted ASR over 80.2% and 44.3%, respectively, while retaining the clean ACC at 82.2%.
>
> ---
>
> Again, we thank the reviewer for the valuable feedback. Please let us know if there are any other questions or suggestions.
>
> Best,
>
> Authors

---

### Meta-Review · Area_Chair_suob · 2023-12-05

**Metareview:**

This paper proposes a way to create a backdoor attack on a diffusion model used for adversarial purification. The diffusion model being used as a preprocessing step in order to ‘remove’ any potential adversarial attack. The backdoor attack uses a trigger pattern to make the diffusion model generate an adversarial example.
On the one hand, the authors have provided extensive experiments demonstrating the validity of their method:
- Transferability
- poisoning during fine-tuning
- On different diffusion models and datasets.

On the other hand, reviewers have noticed that this work only considers the less sophisticated backdoor techniques and does not consider more recent (and advanced) work. Also, the novelty of this paper is diminished by the fact that adversarial purification was already known to be broken (see e.g.,  https://arxiv.org/abs/2211.09565), so backdoors are not needed to break it.
Also, reviewers have questioned the practicability of the method:
- adversarial purification is slower (requires diffusion) and decreases accuracy. So people may not use it in practice
- Poisoning a diffusion model in practice is quite convoluted and expensive.

Overall, I am recommending the rejection of this paper for the reasons above. In particular, the two main reasons are:
- because the setting is quite convoluted---it is assumed an attacker will create a backdoor in a diffusion model that will be used subsequently for adversarial purification.
- And because adversarial purification has been broken without backdoors.

**Justification For Why Not Higher Score:**

There are some concerns around the related work around the type of backdoor attack used (most recent related work could have been considered).

There are some concern regarding the practicability of the threat model (will it happen in practice) and the method (it is feasible in practice)

**Justification For Why Not Lower Score:**

I believe the authors already did a good experimental job and do not need to consider more backdoors/triggers.

Overall I believe that this paper is a good contribution to the current discussion around adversarial purification showing that even that purification process can be attacked.

---

### Decision · Program_Chairs · 2024-01-16

Reject